# A phyB-PIF1-SPA1 kinase regulatory complex promotes photomorphogenesis in Arabidopsis

Inyup Paik [1], Fulu Chen [1,3], Vinh Ngoc Pham[1], Ling Zhu [1,4], Jeong-Il Kim [1,2] & Enamul Huq [1]

CONSTITUTIVELY PHOTOMORPHOGENIC1 (COP1) is a highly conserved E3 ubiquitin ligase from plants to animals and acts as a central repressor of photomorphogenesis in plants. SUPPRESSOR OF PHYA-105 1 family members (SPA1-SPA4) directly interact with COP1 and enhance COP1 activity. Despite the presence of a kinase domain at the N-terminus, no COP1-independent role of SPA proteins has been reported. Here we show that SPA1 acts as a serine/threonine kinase and directly phosphorylates PIF1 in vitro and in vivo. SPAs are necessary for the light-induced phosphorylation, ubiquitination and subsequent degradation of PIF1. Moreover, the red/far-red light photoreceptor phyB interacts with SPA1 through its C-terminus and enhances the recruitment of PIF1 for phosphorylation. These data provide a mechanistic view on how the COP1-SPA complexes serve as an example of a cognate kinase-E3 ligase complex that selectively triggers rapid phosphorylation and removal of its substrates, and how phyB modulates this process to promote photomorphogenesis.

[1] Department of Molecular Biosciences and The Institute for Cellular and Molecular Biology, The University of Texas at Austin, Austin, TX 78712, USA. [2] Department of Biotechnology and Kumho Life Science Laboratory, Chonnam National University, Gwangju 61186, Korea. [3] Present address: MOA Key Laboratory of Soybean Biology (Beijing), National Key Facility of Crop Gene Resource and Genetic Improvement, Institute of Crop Sciences, Chinese Academy of Agricultural Sciences, 100081 Beijing, China. [4] Present address: Syngenta Crop Protection, LLC., Research Triangle Park, NC 27709, USA. Correspondence and requests for materials should be addressed to E.H. (email: huq@austin.utexas.edu)

The ubiquitin-26S proteasome system (UPS) plays critical roles in controlling growth and development of all eukaryotes[1,2] and is involved in multiple developmental and environmental responses, including photomorphogenesis in *Arabidopsis*[2,3]. This system targets specific proteins by selectively attaching a polyubiquitin chain by the action of three enzymes: E1, E2, and E3, and the polyubiquitin chain serves as a tag for specific recognition and degradation of the substrate by the UPS pathway[1,2].

CONTITUTIVE PHOTOMORPHOGENIC 1 (COP1) is a RING finger protein containing WD40 repeats and a coiled-coil domain and is highly conserved from plants to mammals[4]. It is one of the best characterized E3 ubiquitin ligases with broad roles as a central repressor of light signaling in plants to cancer biology in mammals[5–7]. COP1 interacts with multiple substrates either directly or indirectly using accessory proteins and targets them for UPS-mediated degradation. For example, in mammals, COP1 interacts with Tribbles pseudokinases (Trib1-3), and the COP1-Trib complex then recruits multiple substrates for UPS-mediated degradation[8,9]. In plants, COP1 interacts with SUPPRESSOR OF PHYA-105 1 family members (SPA1-SPA4), and the COP1-SPA complexes target multiple transcription factors for rapid degradation[10]. Mutations in *COP1* or *SPA* genes result in either lethality at an early seedling development stage (for *cop1* null alleles) or severe developmental defects including dwarfism and early flowering[11–13]. One characteristic phenotype of the COP1-SPA-related mutants is premature light-induced development even in the absence of light called constitutive photomorphogenic (cop) development[11]. This is because COP1-SPA complex targets positively acting transcription factors (e.g., HY5/LAF1/HFR1 and others) for degradation in darkness. In *cop1* and *spaQ* (*spa1spa2spa3spa4* quadruple) mutants, the stabilized positively acting transcription factors promote photomorphogenic development even in the dark[10].

*SPA1* was originally identified from a genetic screen to isolate a suppressor of a weak phytochrome A mutant *phyA-105*[14,15]. Since then, four *SPA* (*SPA1*, *SPA2*, *SPA3* and *SPA4*) genes have been characterized with differential roles in plant development[13,16,17]. SPAs contain a ser/thr kinase domain at the N-terminus, a coiled-coil domain in the middle, and four WD-40 repeats in the C-terminus that serves as an interaction domain with specific substrates[14]. SPAs directly interact with COP1 through their coiled-coil domain and form multiple COP1-SPA complexes[18]. Genetic evidence shows that SPAs can enhance the COP1 activity in vivo. For example, weak alleles of the *cop1 spa1* double mutant display strongly enhanced photomorphogenic phenotypes in the dark, suggesting SPA1 genetically interacts with COP1[19,20]. In addition, *spaQ* displays a very similar phenotype as a strong allele of *cop1* mutant, suggesting the requirement of SPAs in COP1 activity in vivo[21]. In vitro biochemical assays showed that the SPA1 coiled-coil domain strongly enhances the COP1 E3 ligase activity[19]. In addition, the COP1-SPA complex associates with CULLIN4, and the CUL4[COP1-SPA] promotes degradation of the positively acting transcription factors in the dark to repress photomorphogenesis[22].

In response to environmental light signals, the red/far-red photoreceptors called phytochromes (phy) undergo allosteric changes in conformation (an inactive Pr to a biologically active Pfr form) and migrate into the nucleus[23,24]. The activated phytochromes then interact with the COP1-SPA complex and reorganizes the complex to inhibit the E3 ligase activity[25,26]. COP1 is also excluded from the nucleus in response to light[27–29], and the reduction in COP1 in the nucleus as well as the light-induced inhibition of COP1 activity contribute to the accumulation of the positively acting transcription factors that promote photomorphogenesis in the light[30].

In addition to the inhibition of the COP1-SPA complex, light-activated phytochromes also directly interact with a group of bHLH transcription factors called PHYTOCHROME INTERACTING FACTORs (PIFs)[31,32]. PIFs mainly function negatively in phytochrome signaling pathways. Moreover, PIFs also regulate a wide range of plant responses to light[32,33]. The red-light-activated phytochromes directly interact with PIFs and trigger rapid phosphorylation, ubiquitination and degradation of PIFs to release photomorphogenic development. In this process, multiple kinases and E3 ubiquitin ligases participate in selective removal of PIFs to promote photomorphogenesis[32,34].

Among all PIFs, PIF1 is unique as it is the only PIF that exclusively represses seed germination in the dark[35,36]. PIF1 interacts with the Pfr forms of both phyA and phyB[37], and undergoes the fastest degradation in response to red light with a half-life of <2 min among all PIFs[38]. PIF1 also directly interacts with the COP1-SPA complex, and both PIF1 and COP1-SPA complex repress photomorphogenesis in the dark in a synergistic manner[30,39,40]. However, upon red-light exposure, PIF1 is ubiquitinated by the COP1-SPA complex and is rapidly degraded to allow seed germination to proceed[41]. This selective substrate recognition and ubiquitination by the COP1-SPA complex is promoted by light-induced PIF1 phosphorylation[38]. Casein Kinase 2 (CK2) was previously shown to phosphorylate PIF1 in vitro; however, was not involved in the light-induced phosphorylation of PIF1[42]. Therefore, the protein kinase necessary for the rapid light-induced PIF1 phosphorylation has not been identified.

Similar to the mammalian COP1-associated pseudo-kinase, Trib, plant COP1-associated SPA kinases have no known substrate, although the kinase domain of both Trib and SPAs are necessary for their biological functions[9,43]. Here we provide evidence that SPA1, a component of the COP1-SPA E3 ubiquitin ligase complex itself, acts as a protein kinase. By performing extensive biochemical, genetic and genomic analyses, we provide strong evidence that SPA1 is a bona fide serine/threonine kinase that is necessary for the light-induced phosphorylation of PIF1. We also provide mechanistic details on how the red/far-red light photoreceptor phyB directly recruits SPA1 kinase and this recruitment plays a pivotal role in inducing PIF1 phosphorylation and degradation to promote seed germination and seedling development. These data highlight the importance of COP1-associated kinases not only in enhancing COP1 activity, but also specific phosphorylation of their substrates for rapid ubiquitination and subsequent degradation.

## Results

**SPA1 shows a protein kinase activity in vitro.** Despite the presence of the majority of the signature sequences of a ser/thr kinase-like domain at its N-terminus[14], no kinase activity has been reported for SPA proteins. To examine whether SPA1 can act as a kinase, we purified the GST and strep-tagged N-terminal half of SPA1 (SPA1-Kin) containing the ser/thr kinase domain. The purified SPA1-Kin domain displayed autophosphorylation activity in a protein concentration-dependent manner (Fig. 1a). To test whether the kinase domain is necessary for the kinase activity of SPA1, we created a point mutation in an amino acid (R517E) on the SPA1 kinase domain, which is conserved in SPA1 sequences from multiple plants (Supplementary Fig. 1). This residue is part of a conserved Glu-Arg salt bridge that defines eukaryotic protein kinases[44], and has recently been shown to be critical for its biological function in the dark[43]. We purified both the wild type and the mutant GST-SPA1-Kin proteins and examined their autophosphorylation activities. Results show a significantly reduced auto-phosphorylation activity for the

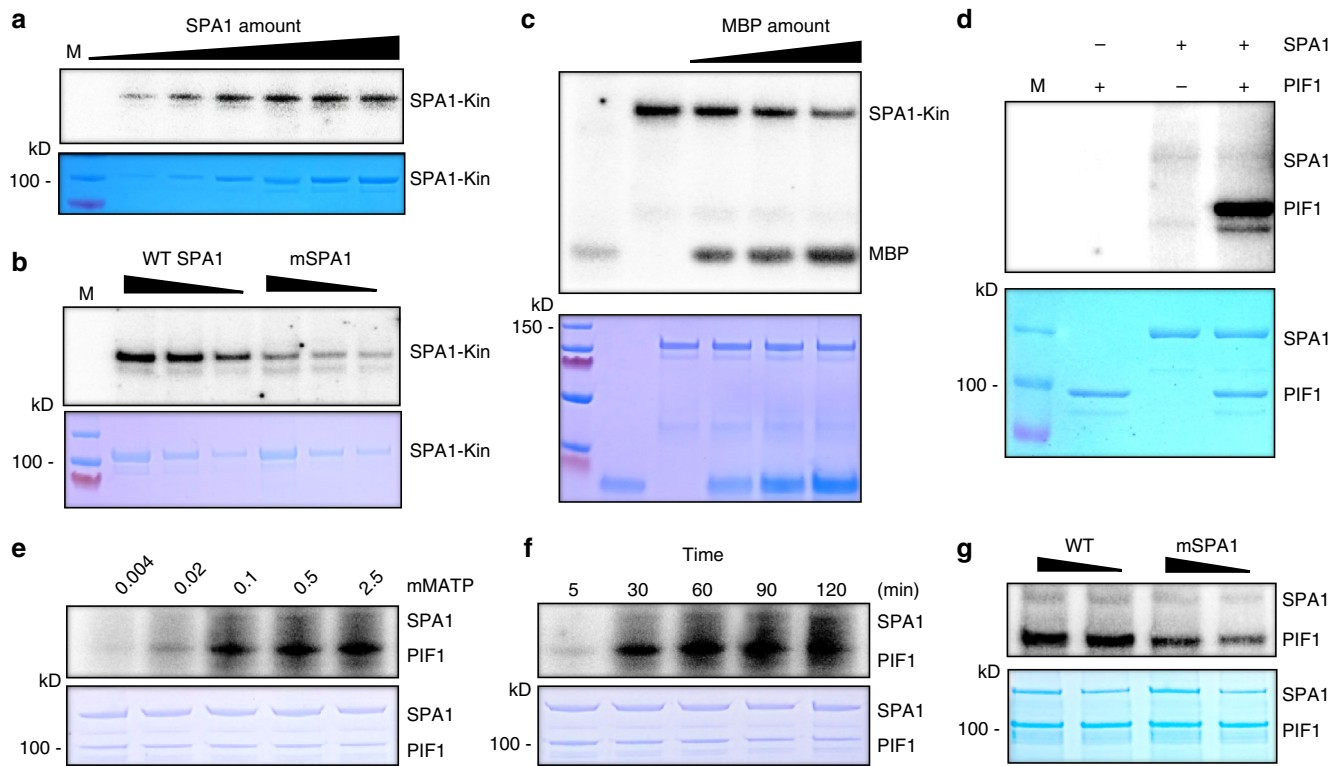

**Fig. 1** SPA1 acts as a ser/thr protein kinase. **a** SPA1 kinase domain (SPA1-Kin) purified from *E. coli* showed an auto-phosphorylation activity in a concentration-dependent manner (autoradiogram on top panel). Bottom panel shows the protein level in a Coomassie-stained gel. M, indicates protein marker. **b** A conserved amino acid mutation on the SPA1 kinase domain reduces the auto-phosphorylation activity of SPA1 (autoradiogram on top panel). Bottom panel shows the protein level in a Coomassie-stained gel. **c** The N-terminal kinase domain of SPA1 exhibits kinase activity in the presence of myelin basic protein (MBP); a general kinase substrate (autoradiogram on top panel). Bottom panel shows the protein levels in a Coomassie-stained gel. **d** Full-length SPA1 purified from *Pichia pastoris* phosphorylates PIF1 in vitro (autoradiogram on top panel). Bottom panel shows the protein levels in a Coomassie-stained gel. **e** ATP-dependent kinase assays of full-length SPA1 on PIF1 (autoradiogram on top panel). ATP concentrations used were 0.004, 0.02, 0.1, 0.5, and 2.5 mM. Bottom panel shows the protein levels in a Coomassie-stained gel. **f** Kinetic analysis of the kinase activity of the full-length SPA1 on PIF1 (autoradiogram on top panel). Bottom panel shows the protein levels in a Coomassie-stained gel. **g** A conserved amino acid mutation on the full-length SPA1 kinase domain reduces the SPA1 kinase activity toward PIF1 (autoradiogram on top panel). Bottom panel shows the protein levels in a Coomassie-stained gel

SPA1-Kin$^{R517E}$ (mSPA1) mutant compared to the wild type SPA1-Kin domain (Fig. 1b), suggesting that the mutation affects the kinase activity of SPA1 in vitro.

To examine whether the SPA1-Kin domain can phosphorylate a substrate, we performed a kinase assay using myelin basic protein (MBP), which has been widely used as a general substrate for many kinases[45]. In vitro kinase assays showed that the SPA1-Kin domain phosphorylates MBP in vitro in a concentration-dependent manner (Fig. 1c). Notably, the SPA1-Kin auto-phosphorylation activity reduces as the MBP phosphorylation increases, showing a general feature of a kinase-substrate relationship in that auto-phosphorylation of a kinase reduces as the substrate phosphorylation increases.

**SPA1 can directly phosphorylate PIF1 in vitro.** Previously, we reported that the COP1-SPA1 E3 ligase can interact with PIF1 in a red light-induced manner and promotes its rapid degradation[41]. To test whether PIF1 could be a native substrate of the SPA1 kinase, a strep-tagged full-length SPA1 protein was purified from a eukaryotic expression host, *Pichia pastoris* (Supplementary Fig. 2a, b), and the kinase assay was performed with PIF1 as a substrate. PIF1 was strongly phosphorylated by SPA1 in vitro (Fig. 1d), suggesting that PIF1 might be a bona fide native substrate of SPA1 kinase. SPA1 also phosphorylated PIF1 in an ATP concentration- and time-dependent manner (Fig. 1e, f). Since

SPA1 harbors a predicted ser/thr kinase domain, we further investigated the nature of PIF1 phosphorylation by acid-base sensitivity assays (Supplementary Fig. 2c). In the presence of KOH, PIF1 was dephosphorylated, while PIF1 phosphorylation was stable in HCl solution, suggesting that SPA1 is a ser/thr kinase for PIF1.

To examine whether the conserved amino acid mutation (R517E) on the full-length SPA1 kinase domain affects PIF1 phosphorylation, we purified the full-length mSPA1 from *Pichia pastoris* and performed the kinase assay using PIF1 as a substrate. The mSPA1 showed significantly reduced PIF1 phosphorylation compared to the wild type SPA1 (Fig. 1g). In addition, the SPA1-Kin domain alone was able to phosphorylate PIF1, while the mutant mSPA1-Kin exhibited reduced PIF1 phosphorylation (Supplementary Fig. 2d). To examine if SPA1 can bind to ATP, we performed a concentration-dependent ATP binding assay using TNP-ATP. Results show that both wild type and mutant form of SPA1-Kin were able to bind to TNP-ATP in a concentration-dependent manner (Supplementary Fig. 2e). Both wild type and mutant SPA1 can directly interact with PIF1 in vitro (Supplementary Fig. 2f, g), suggesting that the reduction in PIF1 phosphorylation by the mSPA1 mutant is not due to a reduction in physical interaction between mSPA1 and PIF1. Taken together, these data strongly suggest that SPA1 is a bona fide ser/thr kinase that phosphorylates PIF1 in vitro.

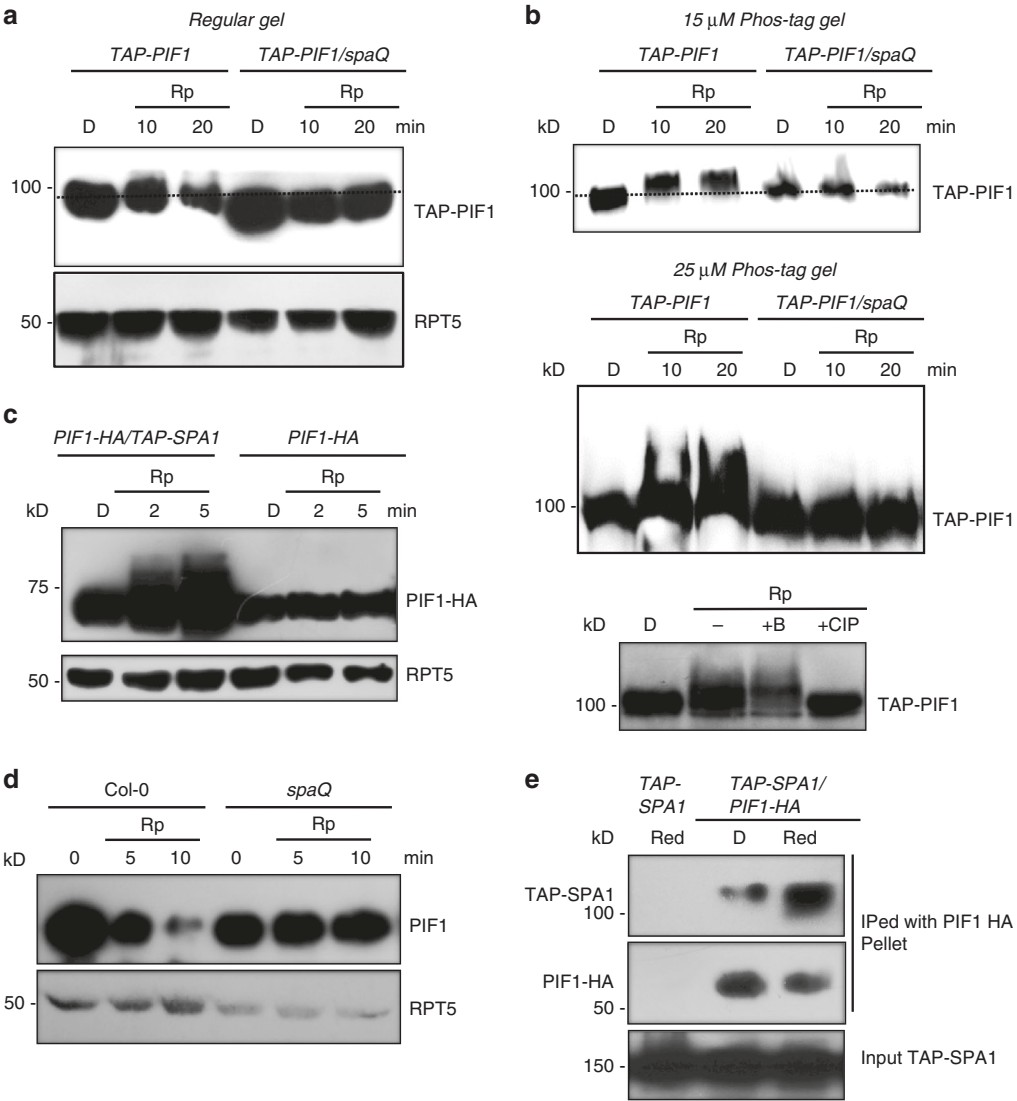

**Fig. 2** SPAs are necessary for the light-induced phosphorylation and degradation of PIF1 in vivo. **a** Immunoblots showing the light-induced phosphorylation of TAP-PIF1 is defective in the *spaQ* mutant compared to wild type. Four-day-old dark-grown seedlings were either kept in darkness or exposed to a pulse of red light (300 µmolm$^{-2}$) and then incubated in the dark for the duration indicated before being sampled for protein extraction. All dash lines show the dark position of the PIF1 band. **b** Immunoblots show a defect in the light-induced phosphorylation of TAP-PIF1 in *spaQ* background compared to wild type in gels containing 15 (top) and 25 (middle) µM phostag. (Bottom panel) The red light-induced slow-migrating band is a phosphorylated form of TAP-PIF1 as indicated by the phosphatase treatment. +CIP, native Calf Intestinal Phosphatase; +B, heat-inactivated boiled CIP. **c** Overexpression of *SPA1* induces faster phosphorylation of PIF1-HA than wild type under weak red-light pulse (1 µmolm$^{-2}$). Seedlings were pretreated with the proteasome inhibitor (40 µM bortezomib) for 5 h before being exposed to red light. Blots were probed with anti-HA and anti-RPT5 antibodies. **d** Immunoblots showing native PIF1 level in *spaQ* mutant compared to wild type. **e** Immunoblots show that TAP-SPA1 can interact with PIF1-HA in a red light-inducible manner in vivo

**SPAs are necessary for light-induced PIF1 phosphorylation.**
Since PIF1 is rapidly phosphorylated in response to light and displays a characteristic mobility shift by immunoblot analysis[38,41], we used *spaQ* plants expressing *pPIF1::TAP-PIF1* to examine the in vivo effect of SPAs on the PIF1 phosphorylation. Strikingly, the rapid red light-induced phosphorylation and consequent band shift of TAP-PIF1 observed in wild type plants was completely abolished in the *spaQ* mutant (Fig. 2a). The PIF1 phosphorylation status was further examined by utilizing Phos-tag containing SDS-PAGE gels (Fig. 2b). In wild type plants, TAP-PIF1 showed clear mobility shift in the presence of 15 and 25 µM Phos-tag. However, in *spaQ* mutant, no mobility shift was observed under these conditions (Fig. 2b, top and middle panel), suggesting a complete absence of the light-induced phosphorylation of PIF1 in vivo. As previously shown[38], the TAP-PIF1

mobility shift was due to the light-induced phosphorylation of PIF1, since treatment with the native calf intestinal phosphatase (CIP) removed the shifted bands from TAP-PIF1 (Fig. 2b, bottom panel). Overexpression of *SPA1* in wild type plants resulted in a faster and more robust phosphorylation of PIF1-HA under weak red-light conditions (Fig. 2c), suggesting an in vivo role of SPA1 as a PIF1 kinase. Taken together, these loss-of-function and gain-of-function analyses suggest that SPAs are essential for the light-induced PIF1 phosphorylation in vivo.

Since the light-induced PIF1 phosphorylation was largely absent in the *spaQ* mutant, we tested if the subsequent PIF1 degradation is also defective. As expected, the PIF1 degradation was almost completely absent in the *spaQ* mutant compared to wild type as previously shown (Fig. 2d). In addition, the light-induced degradation of PIF1 was significantly delayed in the

*spa123* and *spa124* triple mutants compared to wild type (Supplementary Fig. 3a), indicating that SPAs are redundantly functioning in light-mediated PIF1 degradation. In accordance with a previous report[41], the poly-ubiquitination of PIF1 in response to light was largely reduced in the *spaQ* compared to wild type (Supplementary Fig. 3b), confirming a role of the COP1-SPA complex in the light-induced ubiquitination of PIF1. In addition, as previously shown, the in vivo interaction between SPA1 and PIF1 is strongly induced under red light (Fig. 2e). Because PIF1 is phosphorylated in response to red light, the red light-induced PIF1-SPA1 interaction further supports that SPA1 might be recruited as a kinase for the light-induced PIF1 phosphorylation.

Since SPAs are usually present in complexes with COP1 *in planta*[18], and function together with COP1[21], we investigated whether COP1 has any role in the light-induced PIF1 phosphorylation. However, the light-induced PIF1 phosphorylation is not defective in the *cop1-4* mutant (Supplementary Fig. 3c). Since *cop1-4* is a weak allele and still maintains some COP1 activity[12], further experiments are necessary to exclude the role of COP1 in PIF1 phosphorylation.

**SPA1 kinase activity is crucial for seed germination.** PIF1 has been shown to be a master negative regulator of red light-mediated seed germination[35]. As reported previously[41], the *spaQ* mutant exhibited significantly delayed germination in response to light possibly due to the increased stability of PIF1 (Fig. 3a). To examine the biological significance of the SPA1-mediated phosphorylation of PIF1, we expressed both the wild type and the

mutant form of *SPA1* as a fusion protein with Luciferase (LUC) in the *spaQ* background and selected homozygous transgenic plants expressing either the wild type *LUC-SPA1* or the *LUC-mSPA1* in the *spaQ* quadruple background (Supplementary Fig. 10c). Phenotypic analyses showed that the wild type *LUC-SPA1* rescued the seed germination phenotype to levels comparable to the wild type (Fig. 3a). However, the kinase mutant *LUC-mSPA1* failed to complement the light-induced seed germination phenotype of the *spaQ*, supporting a role of the SPA1 kinase activity in regulating the light-induced seed germination through PIF1. At a molecular level, *LUC-mSPA1* plants showed a delayed phosphorylation and subsequent degradation of PIF1 compared to the wild type *LUC-SPA1* in the *spa1spa2spa3* triple background in vivo (Fig. 3b). This is consistent with our biochemical data that mSPA1 has reduced kinase activity; hence, reducing the phosphorylation of PIF1 in vitro and in vivo. As previously reported[43], the *LUC-mSPA1* also largely failed to rescue the constitutive photomorphogenic phenotypes of the *spaQ* in the dark compared to the wild type *LUC-SPA1* (Fig. 3c). These data suggest that the SPA1 kinase function is not only important for regulating seed germination, but also the skotomorphogenic development of Arabidopsis seedlings.

**SPA1 interacts with the C-terminus of phyB under red light.** The red/far-red photoreceptor phyB plays a major role in the red light-mediated PIF1 phosphorylation and degradation in the *phyA* background[38]. To investigate the functional relationship between phyB and SPA1, we first performed yeast two hybrid assays to identify the domains necessary for their interaction. We

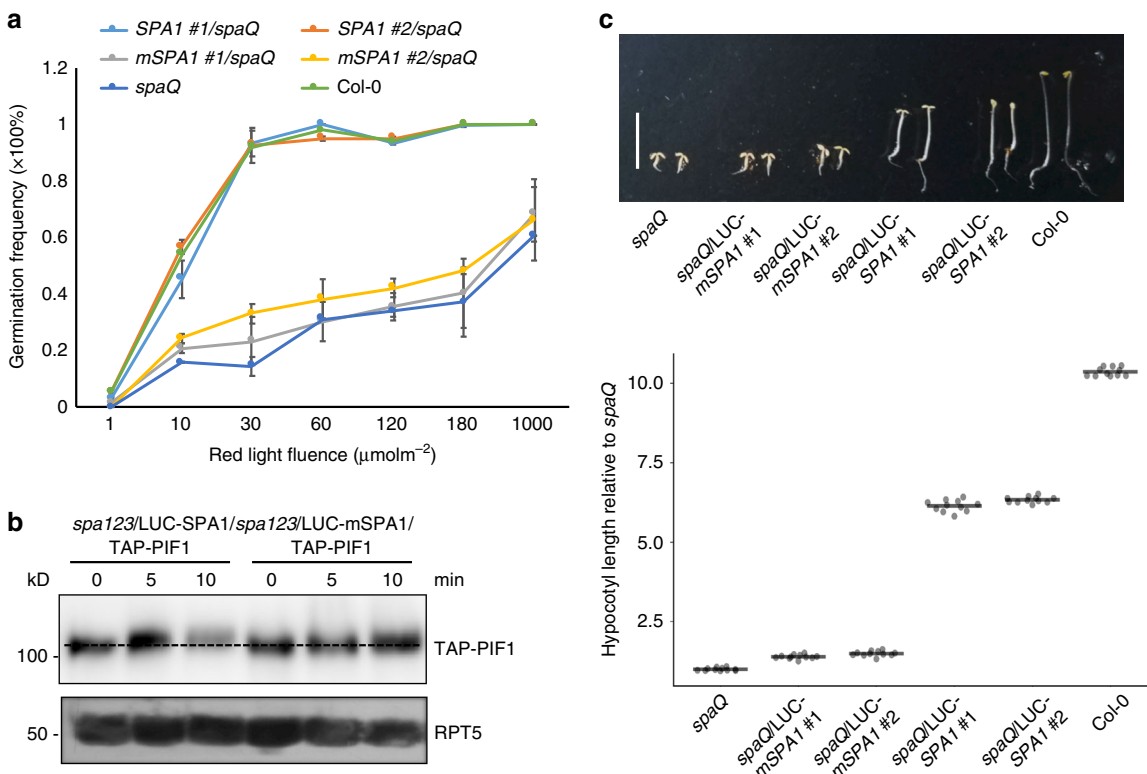

**Fig. 3** The kinase activity of SPA1 is necessary for its biological function. **a** Seed germination phenotypes of wild type and mutant SPA1 expressed in the *spaQ* background in response to an increasing fluence of red light. Col-0 and *spaQ* were used as controls. Homozygous transgenic plants expressing similar levels of wild type and mutant SPA1 were selected. Error bars indicate s.e.m. (*n* = 3). **b** Immunoblots showing the phosphorylation and degradation of TAP-PIF1 in *spa123* seedlings expressing either the wild type or the mutant form of LUC-SPA1. **c** (top) Photograph showing the seedling phenotypes of two independent lines of wild type LUC-SPA1 and LUC-mSPA1 expressed in *spaQ* background grown in darkness for 4 days. The mutant LUC-mSPA1 failed to complement the short hypocotyl phenotype whereas the wild type LUC-SPA1 largely complements the phenotype. (Bottom) Dot plot shows the hypocotyl lengths of seedlings shown in the top panel. Scale bar = 5 mm. Line = median, (*n* = 11)

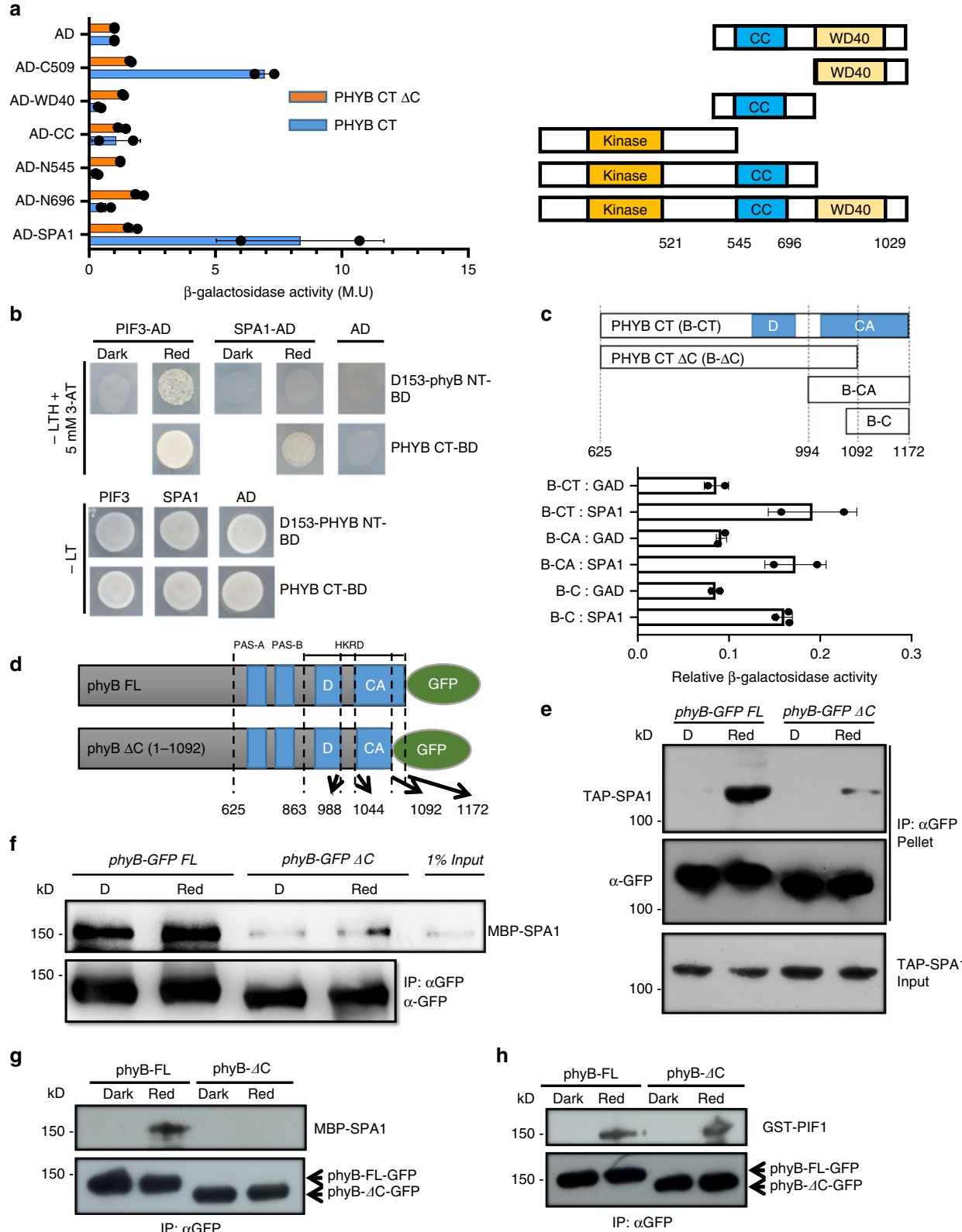

mapped the interaction domain between the coiled-coil (CC) domain and WD-40 repeat (WD40) in the C-terminal half of SPA1, and the C-terminal half of phyB (PHYB CT) using yeast-two-hybrid assays (Fig. 4a). Further deletion analysis showed that the last 80 amino acids of the C-terminal end of phyB is required for the interaction with full-length SPA1 (Fig. 4a). Furthermore, yeast two-hybrid growth assays using SPA1 and the two halves of

phyB showed that that the C-terminal half of phyB, but not the N-terminal half of phyB (NG) interacts with SPA1 in a light-independent manner, while NG interacted with PIF3 under these conditions (Fig. 4b). Conversely, the extreme C-terminal (C) domain of phyB was able to interact with SPA1 and the inter-action level was comparable to the full-length C-terminal PHYB (Fig. 4c). These data are consistent with a previous report[46], and

**Fig. 4** PhyB interacts with SPA1 in a light-dependent manner through its C-terminal domain. **a** Mapping of the interaction domains for phyB and SPA1 using the Yeast-two-hybrid assays. Right panel shows the full-length and various domains of SPA1. Left, β-galactosidase assays show that the Coiled-Coil (CC) domain and WD40 repeat at the C-terminus of SPA1 is necessary for interaction with the C-terminal domain of phyB (PHYB CT, aa 625-1172). On the other hand, the extreme C-terminal 80 amino acids of PHYB-CT is necessary for SPA1 interaction. β-galactosidase activity was normalized by activation domain (AD) empty vector control. LexA-PHYB CT (aa 625-1172), LexA-PHYB CT ΔC (aa 625-1092). Error bar = SD, (n = 2). **b** (Top panel) Yeast-two-hybrid assays in the presence of chromophore phycocyanobillin (PCB). Yeast colonies were grown under red light for two days on –LTH (5 mM 3-AT) drop-out media containing 20 µM of PCB. PIF3 was used as a positive control. The full-length phyB (D153-PHYB) can interact with PIF3 in both dark and red light, while the N-terminal half of phyB (D153-PHYB NT) can only interact with PIF3 under red light. The C-terminal half of phyB also showed interaction with PIF3 independent of light. However, under the same condition both full-length and N-terminal half of phyB failed to show interaction with SPA1, whereas the C-terminal half of phyB showed interaction with SPA1. (Bottom panel) Growth on control plate is normal for all combinations of constructs. **c** The C-terminal 80 amino acids of phyB are sufficient for interaction with SPA1. (Top panel) Schematic diagram of various truncated constructs of phyB fused to BD vector. Full-length SPA1 was fused to AD vector. Error bar = SD, (n = 2). **d** Schematic diagram showing the structures of phyB-GFP FL and phyB-GFP ΔC as a fusion protein with green fluorescent protein (GFP). **e** In vivo co immunoprecipitation assay shows that phyB interacts with SPA1 in response to red light using the last 80 amino acids. phyB-SPA1 interaction is drastically reduced when the putative SPA1-interaction domain in phyB C-terminus was deleted. **f** Semi-in vitro pull-down assay shows that the C-terminal domain of phyB is necessary for SPA1 interaction. **g** In vitro pull-down assay with phyB-GFP purified from yeast and MBP-SPA1 expressed from *E. coli* shows light- and C-terminal-dependent interaction between the two proteins. **h** Integrity of the purified phyB-GFP and phyB ΔC-GFP has been shown by in vitro light-dependent interaction with GST-PIF1

suggest that the putative SPA1 interaction domain at the very C-terminal end of phyB is both necessary and sufficient for SPA1 interaction in yeast. To confirm this hypothesis by an independent method, we cloned the full length PHYB (FL) and phyB ΔC with a GFP tag in a binary vector (Fig. 4d). Stable transgenic plants expressing similar levels of the full-length phyB-GFP FL or phyB-GFP ΔC were generated in the *phyA-211 phyB-9* background (Supplementary Fig. 10a). These lines were crossed with the tandem affinity purification (TAP) tagged SPA1 in *phyAB* background (*phyAB/TAP-SPA1*), and the phyB-SPA1 interaction was tested by a light-dependent in vivo co-immunoprecipitation assay as previously described[26]. Interestingly, the phyB-GFP FL robustly interacted with TAP-SPA1 in a red light-dependent manner in vivo, whereas the phyB-GFP ΔC displayed much reduced interaction with TAP-SPA1 (Fig. 4e). In addition, in a semi-in vitro pull-down assay, we also observed strong interaction between the phyB-GFP FL and MBP-SPA1. However, phyB-GFP ΔC showed much reduced interaction with MBP-SPA1 (Fig. 4f). Because phyB can homo- and hetero-dimerize among other type II phytochromes[47], and these in vivo phyB-GFP FL and phyB-GFP ΔC preparations might have both homo- and hetero-dimers, we performed an in vitro pull-down assay using phyB-GFP FL and phyB-GFP ΔC purified from yeast cells and MBP-SPA1 from *E. coli*. The in vitro pull-down assay showed that phyB-GFP ΔC failed to bind to MBP-SPA1, while phyB-GFP FL strongly interacted with MBP-SPA1 in a light-dependent manner (Fig. 4g). The integrity of the phyB-GFP FL and phyB-GFP ΔC has been examined by the in vitro pull-down assay where both proteins interacted with GST-PIF1 in a light-dependent manner (Fig. 4h). These data collectively suggest that the C-terminal 80 amino acids of phyB are indispensable for the light-induced SPA1 interaction in Arabidopsis.

To rule out the possibility that SPA1 might still interact with the N-terminal domain of phyB, we performed semi-in vivo co-immunoprecipitation assays using the previously described *PBG* (*phyB-GFP FL*), and *NG* (*phyB N terminal-GFP-GUS-NLS*) transgenic plants[48]. Results show that PBG can interact with MBP-SPA1 in a light-stimulated manner (Supplementary Fig. 4), whereas NG failed to interact with MBP-SPA1 under these conditions. Overall, these data suggest that SPA1 interacts with phyB through the C-terminal domain.

**phyB enhances the phyB-SPA1-PIF1 complex formation in vitro.** To address the biochemical relevance of the phyB-SPA1 interaction, we performed an in vitro pull-down assay to test the phyB-SPA1-PIF1 trimolecular complex formation. We expressed

GST-PIF1-strep protein in *E. coli* and examined its interaction with SPA1 in the absence or presence of an increasing amount of phyB. Pull-down using glutathione beads showed an interaction between PIF1 and SPA1 in vitro even in the absence of phyB. However, the PIF1-SPA1 interaction was strongly enhanced in the presence of the Pfr form of phyB (Fig. 5a, b). Moreover, the PIF1-SPA1 interaction was further enhanced when the phyB amount was increased to two-fold. Thus, phyB-SPA1-PIF1 can form a trimolecular complex in vitro.

**Both SPA1 and phyB phosphorylate PIF1 in vitro.** Phytochromes displayed ser/thr kinase activity in vitro[49,50]. Although, the C-terminal half of phytochromes have sequence similarity to histidine kinase-related domains (HKRD), a recent study mapped the kinase activity to the N-terminus of Oat phyA[50]. SPA1 shows kinase activity in vitro, and both phyB and SPA1 are necessary for PIF1 phosphorylation and degradation in vivo. Hence, we examined the effect of phyB on the SPA1-mediated phosphorylation of PIF1 in vitro. We performed a light-dependent in vitro kinase assay to elucidate the molecular effect of phyB on the SPA1 kinase activity and vice versa. Figure 5c shows that both SPA1 and phyB independently phosphorylates PIF1 in vitro. Moreover, PIF1 phosphorylation was strongly enhanced in the presence of both SPA1 and phyB, possibly due to mutual enhancement of PIF1 phosphorylation by each other. However, unlike the in vivo data, no light-induced phosphorylation of PIF1 was observed in vitro. Since the intrinsic kinase activity of Arabidopsis phyB is still debatable, we established a method to purify functional phytochromes from *Saccharomyces cerevisiae*. By using holo-phytochromes, we repeated the in vitro kinase assay for both phyA and phyB. In this assay, both phyA and phyB phosphorylated PIF1 in vitro (Supplementary Fig. 5a). The quality of these phytochrome preparations has been examined by zinc blot (Supplementary Fig. 5a), and in vitro interaction with GST-PIF1 in a light-dependent manner (Supplementary Fig. 5b). These data support the hypothesis that phytochromes might function as protein kinases for PIFs.

**phyB-SPA1 interaction is necessary for PIF1 phosphorylation.** To elucidate the biological relevance underlying the phyB-SPA1 interaction, we examined PIF1 levels in both wild type *PHYB-GFP FL* and *PHYB-GFP ΔC* transgenic plants. The light-induced degradation of PIF1 is strongly reduced in the *PHYB-GFP ΔC* lines compared to *PHYB-GFP FL* lines (Fig. 6a). In the *PHYB-GFP ΔC* lines, even after 2 hours of red light exposure, the PIF1

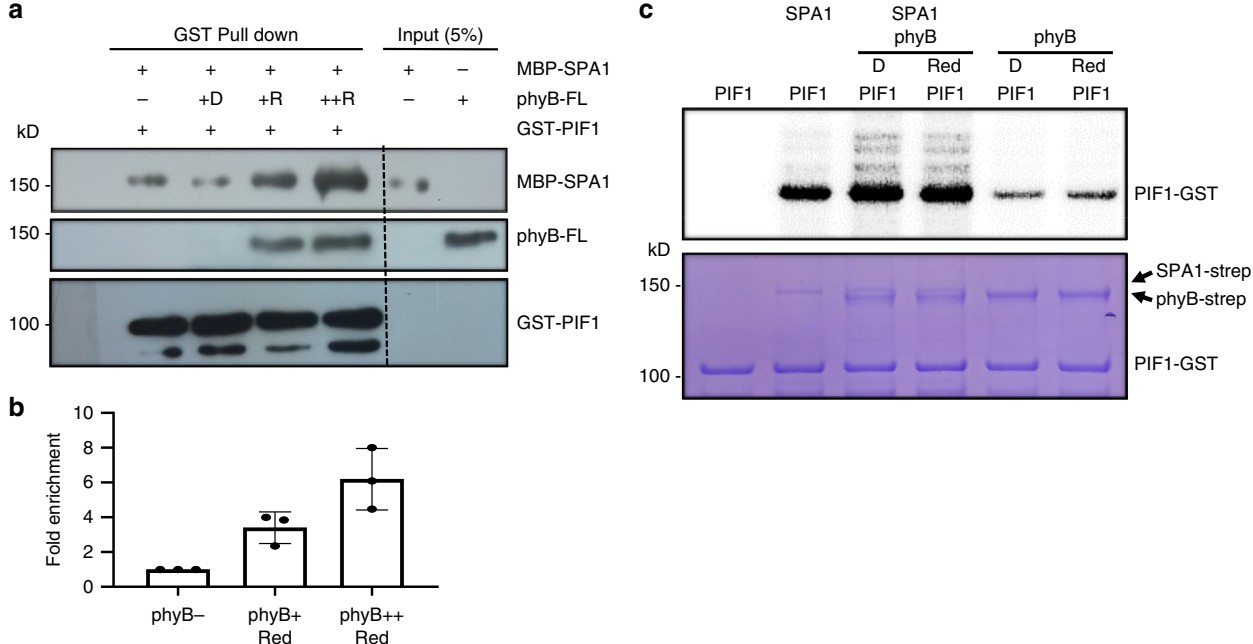

**Fig. 5** PhyB interaction with SPA1 promotes phyB-SPA1-PIF1 trimolecular complex formation and enhanced phosphorylation of PIF1 in vitro. **a** In vitro pull-down assay shows that phyB enhances PIF1-SPA1 interaction in a light- and concentration-dependent manner. **b** Bar graph shows the enhanced interaction between SPA1 and PIF1 in the presence of phyB. Error bar = SD, (n = 3). **c** (top panel) Autoradiogram shows that phyB enhances SPA1-mediated phosphorylation of PIF1 in vitro. (Bottom panel) Coomassie-stained gel shows the amount of various proteins used in the assay

level remained at ~60% of the dark level (Fig. 6a, b). However, the *PHYB-GFP FL* lines showed complete PIF1 degradation in <30 min of red light, similar to wild type (Fig. 6a, b). We also examined the phosphorylation status of PIF1 using Phos-tag and regular SDS-PAGE gels. In these assays, TAP-PIF1 displayed no band-shift in response to red light in the *PHYB-GFP ΔC* plants, while the wild type *PHYB-GFP FL* induced a band-shift of TAP-PIF1 under these conditions (Fig. 6c, d), indicating an absence of phosphorylation in the *PHYB-GFP ΔC* transgenic plants. These data strongly suggest that the phyB-SPA1 interaction is essential for phyB to recruit SPA1 into a phyB-SPA1-PIF1 trimolecular complex and facilitate PIF1 phosphorylation and degradation in response to light.

**phyB C-terminal domain is essential for photobody formation**. phyB has been shown to form nuclear photobodies in response to red light, and the size and number of photobodies have been implicated to phyB function[24,51,52]. To examine whether the C-terminal domain of phyB is necessary for the photobody formation, we grew *PHYB-GFP FL* and *PHYB-GFP ΔC* transgenic seedlings in the dark and then exposed them to red light for various time periods. *phyB-GFP FL* seedlings displayed normal nuclear photobodies as previously described[51]. However, *phyB-GFP ΔC* transgenic seedlings failed to form nuclear photobodies even after prolonged light exposure (Fig. 6e, Supplementary Fig. 6). To examine if the C-terminal domain is necessary for phyB nuclear transport, we measured the cytosolic GFP signal in the hypocotyl cells of both *PHYB-GFP FL* and *PHYB-GFP ΔC* transgenic seedlings. Results show that the GFP signal is much more abundant in the cytoplasm of *PHYB-GFP ΔC* compared to *PHYB-GFP FL* seedlings (Fig. 6f). These data are consistent with a previous report that phyB-NG-NLS which lacks the C-terminal half of phyB failed to form nuclear photobodies under red light. In contrast, the C-terminal domain of phyB is sufficient to form nuclear bodies even in darkness[48]. Thus, these results suggest that the interaction between phyB and SPA1 and possibly other

proteins including other SPAs through the C-terminal domain might be necessary for robust phyB nuclear transport and photobody formation in response to light.

**phyB C-terminus is necessary for multiple phytochrome responses**. To examine the biological significance of phyB-SPA1 interaction, we examined a variety of phenotypes of *PHYB-GFP FL* and *PHY-GFP ΔC* transgenic plants compared to wild type (Fig. 7a). Because of PIF1's exclusive role in light-regulated germination, we first examined the seed germination phenotypes of these genotypes. As shown in Fig. 7b, *PHYB-GFP ΔC* seeds showed significantly delayed germination compared to *PHYB-GFP FL*, which is similar to *spaQ* seeds (Fig. 3a), again supporting the role of the SPA1-interaction domain of phyB in regulating seed germination in response to red light. Previous genetic screens have described the isolation of multiple *phyB* hyposensitive mutants[53]. Among them, *phyB-28* mutant has a premature stop codon resulting in the production of phyB with similar deletion (1-991) to phyB-GFP ΔC (1-1092) (Fig. 7a)[54]. To examine whether *phyB-28* shows similar germination defects as observed in *PHYB-GFP ΔC*, we generated transgenic plants expressing *PHYB-28* in the *phyA-211phyB-9* background similar to a wild type *phyA-211* level (Supplementary Fig. 10b). Compared to wild type (*phyA-211*), *phyB-28* showed a significant delay in germination rate in response to red light (Fig. 7c), suggesting that the hyposensitive phenotypes of the *phyB-28* allele might be due to a defect in the interaction with SPA proteins. Consistent with the above genetic analyses, the light-induced phosphorylation and degradation of PIF1 is also defective in *phyB-28* allele compared to *phyA211* control (Fig. 7d).

In addition to seed germination, other phytochrome-mediated responses are also significantly reduced in the *phyB-GFP ΔC* and *phyB-28* mutants compared to wild type *phyB-GFP FL*. Hypocotyl lengths under different fluences of red light were measured for *phyB-GFP FL* and *phyB-GFP ΔC* as well as *phyB-28* expressing plants. Results show that the hypocotyl lengths of *phyB-GFP ΔC*

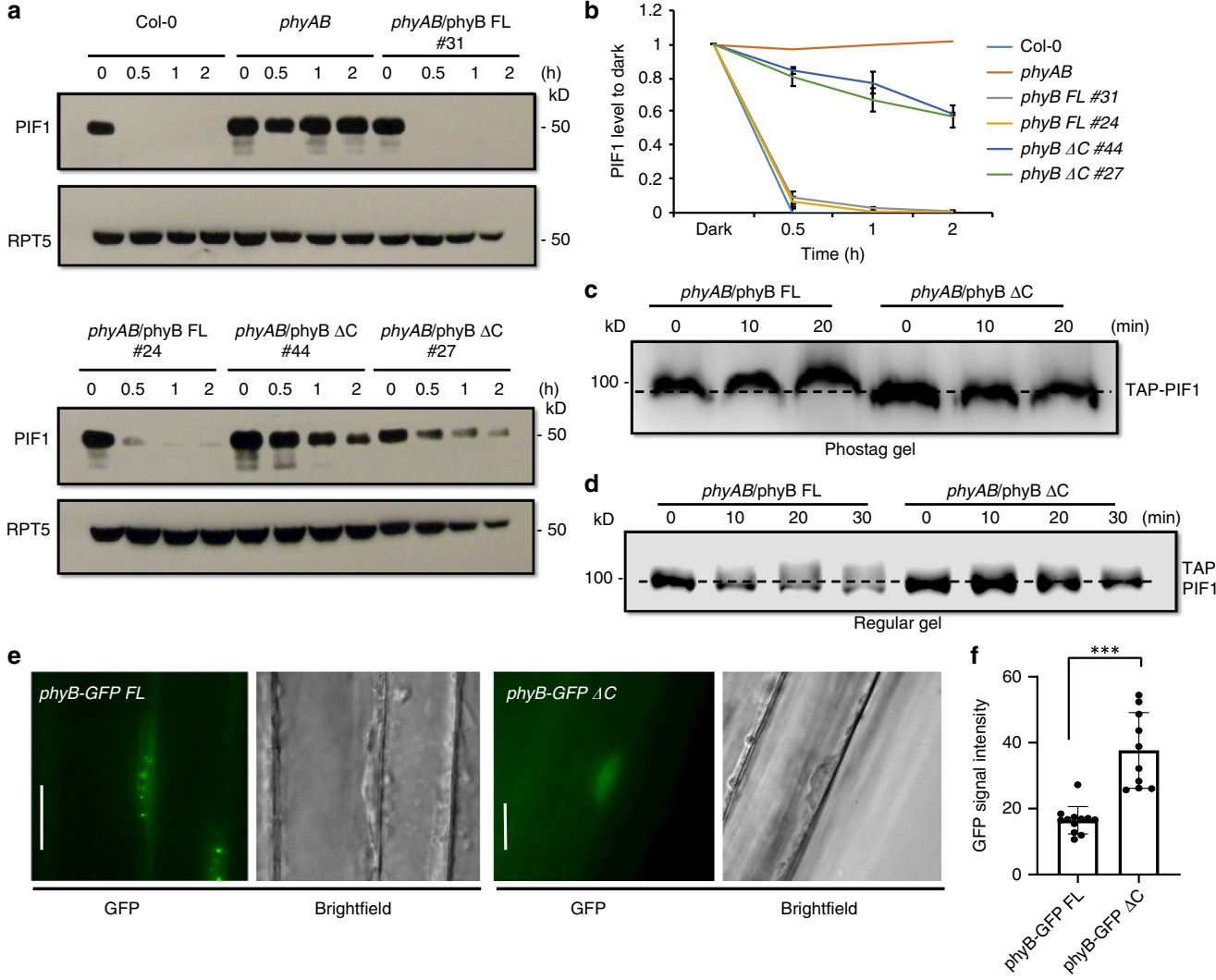

**Fig. 6** The C-terminal SPA1-interaction domain of phyB is essential for photobody formation and PIF1 phosphorylation. **a** Immunoblots showing the level of native PIF1 in two independent transgenic lines of phyB-GFP FL (#31 and #24) and phyB-GFP ΔC (#44 and #27) along with wild type and *phyAB* as controls. Total protein was extracted from 4-day-old dark-grown seedlings either kept in darkness or exposed to continuous red light (3.5 µmolm$^{-2}$ s$^{-1}$) over time. **b** A graph showing the amount of PIF1 levels in response to red light exposure over time. **c**, **d** The light-induced phosphorylation of PIF1 is defective in phyB-GFP ΔC compared to phyB-GFP FL. **e** The phyB-SPA1 interaction is necessary for phyB nuclear transport and photobody formation. The phyB-GFP FL produces nuclear photobodies after 5 h of red light (10 µmolm$^{-2}$ s$^{-1}$) treatment (left), whereas phyB-GFP ΔC showed dispersed nuclear signals (right). Bar = 15 µm. **f** The cytoplasmic GFP signals were quantified from phyB-GFP FL and phyB-GFP ΔC images by imageJ ($n = 12$), Error bar = SD. ***$p$-value < 0.00001, student's $t$-test

and *phyB-28* are significantly longer compared to controls under various fluences of red light (Supplementary Fig. 7a, b, c). Measurement of the petiole lengths using 3-week-old plants show that *phyB-GFP ΔC* has significantly longer petiole lengths compared to those of *phyB-GFP FL* (Supplementary Fig. 7d), suggesting a role of the phyB C-terminus in multiple phytochrome responses. Consistent with our results, the previously reported *NG* and *CG* seedlings showed more stable PIF1 under red light than *PBG* (Supplementary Fig. 7e); although, both *NG* and *CG* induced a slight degradation of PIF1 compared to *phyAphyB*. Recently, *CG* has been shown to induce degradation of PIFs[55]. Because NG undergoes rapid degradation compared to PBG[56], and NG and the isolated N-terminal photosensory domain can interact with PIFs[57,58], it is possible that NG undergoes co-degradation with PIF1 through PPK kinases or additional unknown kinases[59]. Taken together, by analyzing multiple *phyB* mutants, these data strongly support the

hypothesis that the extreme C-terminal domain of phyB is necessary for many phyB-mediated responses during photomorphogenesis.

Several lines of evidence show that the C-terminal end of phyB is specifically necessary for SPA1 interaction and PIF1 degradation, as opposed to a defect in interaction with PIF1 and/or lack of dimerization. As shown in Supplementary Fig. 8a, PIF1 interacts with both phyB-GFP FL and phyB-GFP ΔC with a similar strength in vivo. Second, an in vitro native PAGE gel shows that phyB-GFP ΔC can dimerize in a similar manner to the phyB-GFP FL protein (Supplementary Fig. 8b). However, the isolated phyB CT ΔC failed to dimerize in yeast assays, despite similar dimerization activity of the phyB CT ΔC with phyB CT in yeast two-hybrid assays (Supplementary Fig. 8c), suggesting that the dimerization might be defective in the isolated phyB CT ΔC, but not in the phyB-GFP ΔC context. Third, chromatin immunoprecipitation (ChIP) assay showed that phyB-GFP FL,

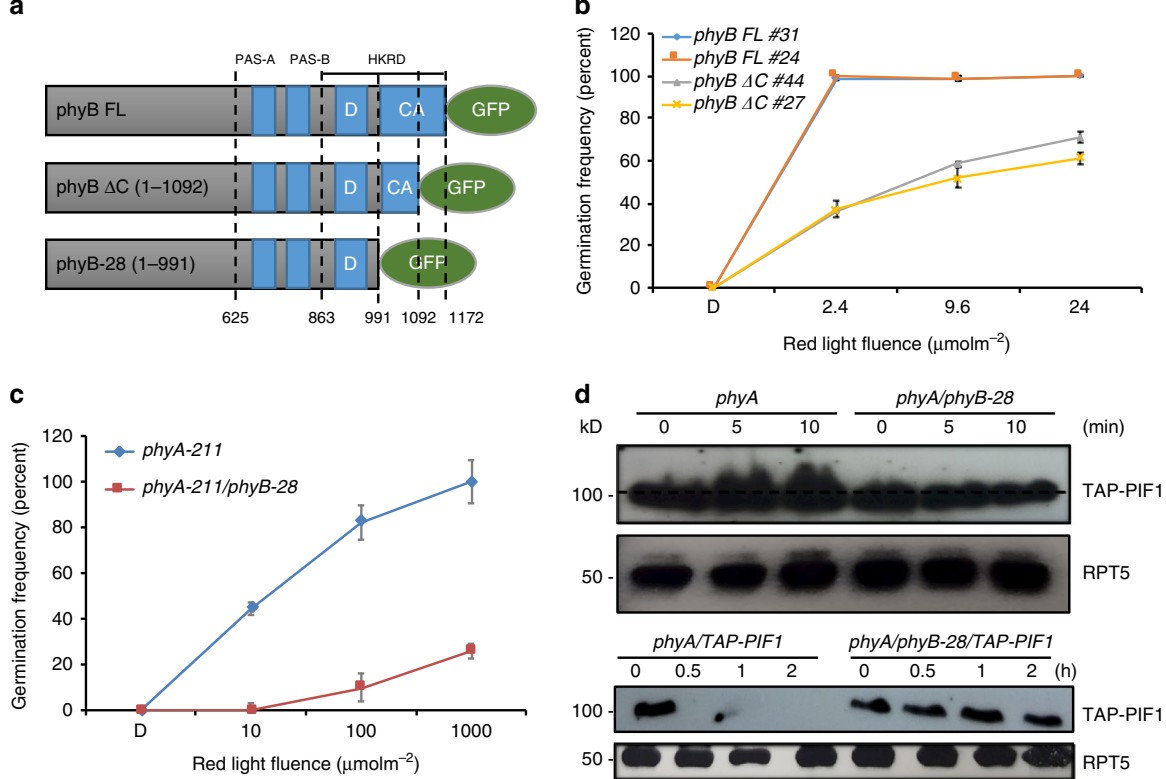

**Fig. 7** Direct interaction between phyB and SPA1 is necessary for phyB-mediated seed germination. **a** Schematic diagram of phyB-28, phyB-GFP ΔC and phyB-GFP FL proteins. **b**, **c** The C-terminal SPA1-interaction domain of phyB is necessary for phyB-mediated seed germination. *phyB-GFP ΔC* displays reduced germination compared to *phyB-GFP FL* lines, while *phyB-28* displays slower germination compared to wild type (*phyA-211* that contains wild type *PHYB*). The error bars indicate s.e.m. (n = 3). **d** The light-induced phosphorylation and degradation of TAP-PIF1 is defective in *phyB-28* compared to wild type (*phyA-211* that contains wild type *PHYB*). Total protein was extracted from four-day-old dark-grown seedlings either kept in darkness or exposed to continuous red light (3.5 μmolm$^{-2}$s$^{-1}$) over time

as well as phyB-GFP ΔC equally sequester PIF1 from the G-box on the *PIL1* promoter in response to red light (Supplementary Fig. 8d). Fourth, we introduced mutations in the four conserved amino acids in the HKRD domain to test whether the possible HKRD-related activity was affected in phyB-GFP ΔC (Supplementary Fig. 8e), and made homozygous transgenic plants expressing the wild type phyB-GFP FL and phyB-GFP 4 M in the *phyA211-phyB-9* background. Immunoblots show that the red light-induced PIF1 degradation was not affected in the *phyB-GFP 4 M* compared to *phyB-GFP FL* (Supplementary Fig. 8f), suggesting that the residues within the HKRD-related domain are not necessary for PIF1 degradation. Finally, we purified phyB-GFP FL, phyB-GFP ΔC, and phyB-GFP 4 M proteins from *Saccharomyces cerevisiae* and confirmed no change in the intrinsic phyB kinase activities in these mutants (Supplementary Fig. 8g). Because phyB-GFP ΔC failed to form nuclear photobodies under red light and also defective in SPA1 interaction, the light-induced degradation of PIF1 might be promoted by either phyB-SPA1 interaction and/or phyB photobody formation in Arabidopsis.

**SPAs are necessary for PIF-target gene regulation**. To understand the molecular basis of SPA- and phyB-mediated regulation of PIF1 stability, we also performed RNA-seq analysis for *spaQ*, *cop1-4* and Col-0 using 3-day-old dark-grown seedlings and dark-grown seedlings exposed to red light for 1 h. In wild type Col-0, 2702 genes are differentially expressed between dark and 1 h of red light treated samples (Fig. 8a), which are categorized as red light responsive genes in the wild type. Red light responsive genes between 1 h of red light treatment compared to dark in *cop1-4*

and *spaQ* are also presented in Supplemental Data 1. In *spaQ* and *cop1-4* mutants, however, 1861 genes are no longer responding to the red-light treatment, suggesting the importance of COP1-SPA complex in regulating red light-responsive gene expression (Supplemental Data 2). Remarkably, the *spaQ* mutant showed 854 genes that do not overlap with *cop1-4* differentially expressed genes (Fig. 8a, Supplemental Data 3). The existence of these exclusively SPA-dependent genes supports a COP1-independent function of SPA proteins.

We also performed RNA-seq analysis for *phyB-GFP FL* and *phyB-GFP ΔC* under the same conditions. A total of 1117 genes were differentially expressed in *phyB-GFP FL* in response to red light within 1 h (Fig. 8b, Supplemental Data 1, 4). Among them, 561 genes were misregulated in *phyB-GFP ΔC* mutant that did not display light responsiveness (Supplemental Data 5). Thus, we categorized these genes into phyB C-terminal-dependent red-responsive genes (Fig. 8b). By comparing the differential gene expression in Col-0 Red/Dark and *spaQ* Red/Dark (Fig. 8a), 2199 genes no longer responding to red light treatment in *spaQ* and are defined as SPA-dependent red responsive genes. Because we identified the phyB C-terminus as a SPA1 interaction domain (Fig. 4), we compared the list of phyB C-terminal-dependent red-responsive genes with the SPA-dependent red-responsive genes. These analyses showed that ~59% (333/561) of the phyB C-terminal-dependent genes also overlap with SPA-dependent genes (Fig. 8c, Supplemental Data 6), suggesting a common regulatory pathway shared by phyB and SPA proteins. Red light responsive *spaQ*-regulated genes were identified as differentially expressed genes between *spaQ* in red light compared to Col-0 in

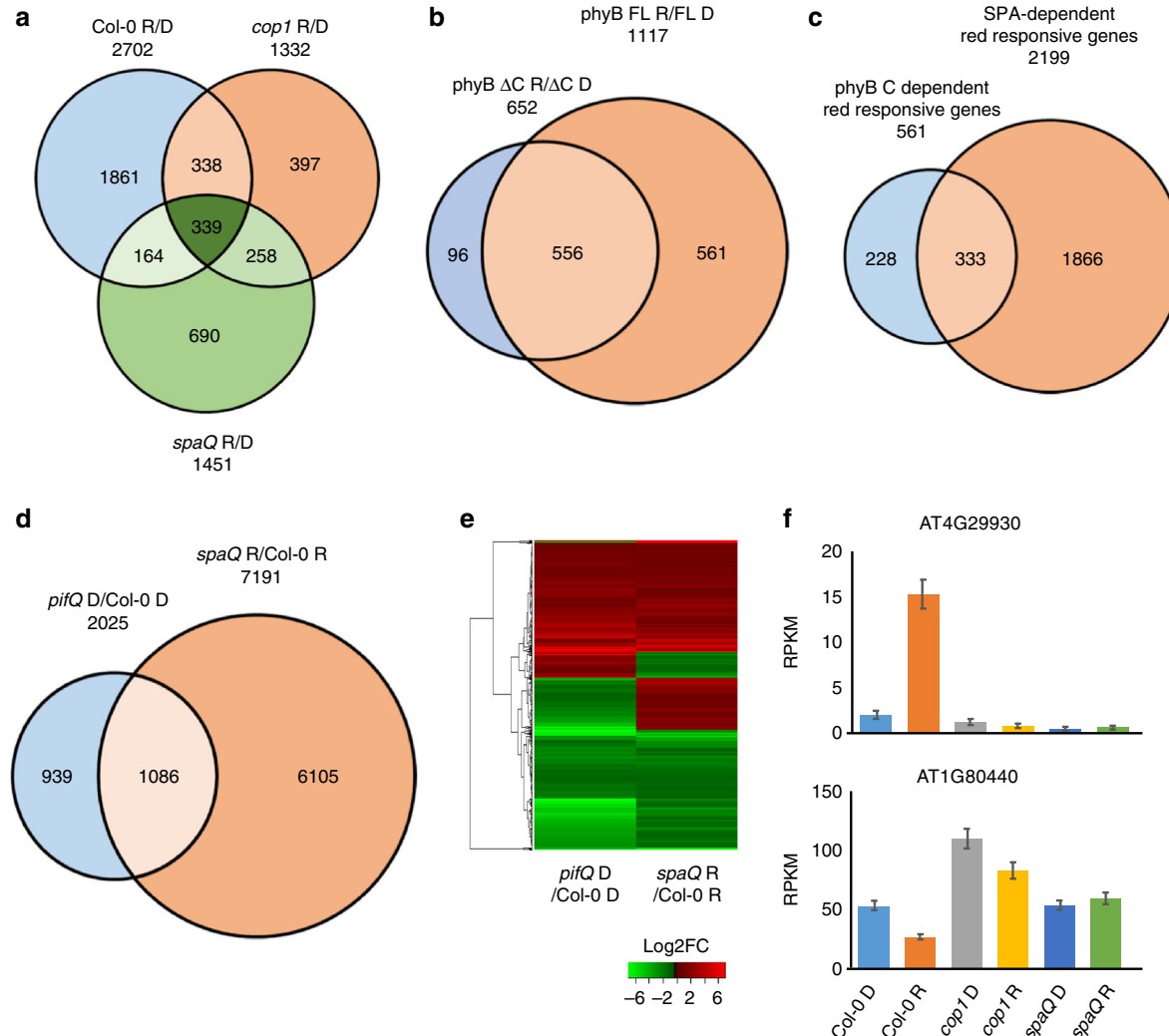

**Fig. 8** RNA-sequencing revealed unique roles of SPAs in PIF-regulated gene expression. **a** Venn diagram of differentially expressed genes (DEGs) in a three-way comparison as indicated (Col-0 R/D, *spaQ* R/D and *cop1-4* R/D). Genes showing differential expression in red light vs dark in three genotypes are presented. **b** Venn diagram of DEGs in phyB-GFP FL vs phyB-GFP ΔC in response to red light. A total of 561 genes was identified as DEGs in phyB-GFP FL that did not show differential expression in phyB-GFP ΔC. **c** Venn diagram showing DEGs in a pairwise comparison between SPA-dependent red light responsive genes and phyB C-terminal-dependent red light responsive genes. Sixty percent of phyB C terminal-dependent red light responsive genes are overlapped with SPA-dependent red responsive genes, supporting a role of the phyB C-terminal domain in SPA-regulated gene expression in response to red light. **d** Venn diagram showing DEGs in a pairwise comparison between *pifQ*-dependent genes in darkness and *spaQ*-dependent red light responsive genes. Approximately, 54% of the PIF target genes are mis-regulated in *spaQ* mutant. **e** Hierarchical clustering of 1086 DEGs in two different pairwise comparisons as indicated (*pifQ* D/Col-0 D and *spaQ* R/Col-0 R). **f** Bar graph showing expression level of one light-inducible and one light-repressive gene in three genotypes as indicated. Reads Per Kilobase of transcript per Million (RPKM) values were obtained from RNAseq data. Error bar = SD, (*n* = 3)

red light (*spaQ* Red/ Col-0 Red) (7191 genes). PIF-mis-regulated genes (2025 genes) in darkness were previously described[60]. We found ~54% (1086/2025) of the PIF-mis-regulated genes in darkness overlapped with the *spaQ*-regulated genes in response to red light (Fig. 8d, Supplemental Data 7). We generated a heatmap for these 1086 overlapping genes and compared their gene expression changes in two different genotypes (*pifQ* vs *spaQ*). A large number of genes showed a similar pattern of up (red)- or down (green)-regulation compared to wild type (Fig. 8e), suggesting that SPA-regulated genes in early red light are largely PIF-regulated as well. Among these genes, Fig. 8f shows an example of the expression level for one light-inducible and one light-repressed gene in *spaQ* and *cop1-4* in comparison to wild type. Moreover, we selected a few of these genes and verified their expression levels by an independent method using RT-qPCR assays. Results show that the RNA-seq data are largely

reproducible (Supplementary Fig. 9). Taken together, the RNA-seq data further supports the proposed trimolecular framework among phyB-SPA-PIF1 in regulating gene expression and photomorphogenesis in response to red light.

## Discussion
Substrate recognition by E3 ubiquitin ligases and subsequent degradation is one of the most critical cellular processes that is regulated by multiple signaling cascades in plants and animals[2,61]. Specific recognition of a substrate by an E3 ubiquitin ligase can be enhanced by phosphorylation of a substrate which increases the affinity between the substrate and the E3 ligase[1,62]. To promote photomorphogenesis, PIFs are phosphorylated, ubiquitinated and then degraded under light[31,32]. This study provides a mechanistic view on how PIF1 is phosphorylated and ubiquitinated by a

cognate kinase-E3 ubiquitin ligase (COP1-SPA) for rapid light-induced degradation.

Although, initial studies on the SPA proteins were focused on their accessory role in enhancing COP1 E3 ligase function, a COP1-independent role for SPAs has not been shown yet, despite the presence of a kinase-like domain at the N-terminus. Here we show that the N-terminal domain of SPA1 possesses a ser/thr kinase activity. Recombinant full-length SPA1 and the N-terminal kinase-like catalytic domain (SPA1-Kin) display autophosphorylation activity (Fig. 1). Both proteins also phosphorylate a generic substrate MBP and a native plant substrate PIF1 in a concentration-dependent manner in vitro. The SPA1-mediated phosphorylation is acid stable and base labile, suggesting that the phosphorylated residues are serine and/or threonine (Supplementary Fig. 2c). Mutation in a critical residue of SPA1 (R517E) abrogating a conserved Glu-Arg salt bridge essential for eukaryotic protein kinases results in reduced kinase activity toward MBP and PIF1 in vitro (Fig. 1). The wild type and mutant SPA1-Kin domains equally bound to an ATP analog (TNP-ATP) in a concentration-dependent manner in vitro (Supplementary Fig. 2e), suggesting that the mutant SPA1-Kin is not defective in ATP binding. Consistently, the light-induced phosphorylation of PIF1 is reduced in the mutant SPA1 compared to wild type SPA1 transgenic plants (Fig. 3c). Moreover, the light-induced phosphorylation of PIF1 is drastically reduced in spaQ compared to wild type (Fig. 2). In addition, the mutant SPA1 failed to rescue the seedling de-etiolation phenotype in the dark and the light-induced seed germination phenotype compared to wild type SPA1 transgenic plants (Fig. 3a, b). Taken together, these data strongly support the conclusion that SPA1 acts as a bona fide ser/thr kinase for PIF1.

Recently, it was shown that SPA1[R517E] mutant failed to rescue the seedling deetiolation phenotype of spaQ[43], and these authors hypothesized that the kinase domain of SPA1 may provide structural information critical for SPA1 function. We have reproduced these data independently and also show additional phenotypes defective in this mutant. Our data show that the kinase domain displays a protein kinase activity in phosphorylating PIF1 in vitro and in vivo. Moreover, the kinase like domain of SPA1 may act as a molecular scaffold for potential protein-protein interaction. The arginine 517 in SPA1 is very well conserved among many plant species, suggesting an importance of the structural integrity of the kinase domain (Supplementary Fig. 1). It is possible that the SPA1[R517E] mutant compromised not only the kinase activity, but also the structural integrity of the SPA1 kinase domain, resulting in a defect in PIF1 degradation and seed germination in response to light. The biochemical basis for the failure of the SPA1[R517E] mutant to rescue the spaQ seedling de-etiolation phenotype in the dark is still unknown. However, our hypothesis is that the kinase activity by itself and/or the structural information included within the kinase domain contribute to regulating seedling de-etiolation phenotypes.

Direct interaction of PIF1 with the Pfr form of phyB is necessary for the rapid light-induced phosphorylation and degradation of PIF1[38]. The in vitro and in vivo data presented here show that phyB is promoting the PIF1-SPA1-phyB complex formation in a red light-inducible manner. Domain mapping analyses shows the last 80 amino acids at the C-terminal end of phyB is necessary and sufficient for SPA1 interaction (Fig. 4a–e). Deletion of the SPA1 binding domain from phyB (phyB ΔC) results in reduced PIF1 phosphorylation and degradation and hyposensitive phenotypes of phyB-GFP ΔC compared to wild type phyB-GFP FL plants (Figs. 6, 7, Supplementary Fig. 7). In addition, three other phyB C-terminal deletion mutants, phyB-28, CG and NG, that were previously reported[48,54], also showed reduced phosphorylation and degradation of PIF1 similar to that

of phyB-GFP ΔC (Fig. 7, Supplementary Fig. 7), further supporting that the C-terminal end of phyB is required for phyB-SPA1 interaction. Overall, these data strongly suggest that the recruitment of SPA1 kinase in a PIF1-SPA1-phyB trimolecular complex is a critical step for the rapid light-induced phosphorylation of PIF1.

The importance of the C-terminal domain of phytochromes in mediating light signaling has been in focus for decades[63]. Recently, Qiu et al showed that overexpression of the C-terminal domain of phyB (625 -1172 aa) induces degradation of PIF3 and activates light-inducible gene expression in the dark[55]. Consistent with this study, Park et al (2018) showed that phyB regulates PIF activity by two independent methods: PIF sequestration, which is induced by the N-terminal half of phyB[57,64], and PIF degradation, which is induced by the C-terminal half of phyB[64]. These two separable activities of phyB are located in two halves of phyB protein. Although, these data highlight the importance of the C-terminal domain of phyB in regulation of PIF abundance, our data show that the last 80 amino acids of the C-terminal domain of phyB is necessary for the light-dependent SPA1 interaction both in vitro and in vivo. Moreover, this domain is necessary for phyB nuclear translocation and photobody formation in response to light (Fig. 6e). SPA proteins co-translocate with phyB into the nucleus[46]. However, it is highly unlikely that SPA proteins are the only factors facilitating transport of phyB into nucleus, as PIFs also promote nuclear translocation of phyB[65]. It is possible that this domain interacts with SPA proteins as well as other factors, and these combined interactions are required for phyB nuclear transport and photobody formation. Because phyB-GFP ΔC behaves normally in terms of chromophore binding, dimerization, PIF1 interaction and sequestration, and complements the phyB phenotypes to a large extent, the defects observed in phyB-GFP ΔC seedlings might not be due to a non-functional protein, rather due in part to a lack of phyB-SPA1 binding in addition to other potential unknown interactions necessary for robust phyB nuclear transport and photobody formation. However, it is still possible that phyB-ΔC might have additional intrinsic defects that were not fully addressed in our study. For example, the phyB-GFP ΔC exhibits defect in the photobody formation similar to the PBY18-YFP (D1040V), which cannot dimerize[55]. But unlike PBY18-YFP (D1040V) which lacks PIF3 interaction, phyB-GFP ΔC still interacts with PIF1 in vivo. Because the isolated phyB CT ΔC failed to dimerize in yeast, it is still possible that the last 80 amino acids of phyB contributes to the dimerization and photobody formation of phyB-GFP ΔC.

Multiple kinases phosphorylate PIFs in vitro including, phytochrome itself[32,42,49,50,59,66]. However, Casein Kinase II (CK2) and BRASSINOSTEROID- INSENSITIVE2 (BIN2) were shown to phosphorylate PIF1 and PIF4, respectively, only in vitro in a light-independent manner. Photoregulatory Protein Kinases (PPK1–PPK4) phosphorylate PIF3 both in vitro and in vivo, and the light-induced phosphorylation of PIF3 is defective in ppk mutants compared to wild type[59]. However, PPKs do not interact with phyB in a light-dependent manner in vitro. Moreover, PPK-mediated phosphorylation of PIF3 might be involved in signal attenuation as opposed to signal transduction, as the ppk mutants display hypersensitive phenotypes in response to red light. A recent study also showed that phytochrome itself acts as a kinase for PIFs[50]. However, the kinase activity of oat phyA has been mapped to the N-terminus as opposed to a conserved HKRD located at the C-terminus. Although the location of the kinase domain in Arabidopsis phys is still being investigated, purified Arabidopsis phyA and phyB from two different expression hosts induce phosphorylation of PIF1 in vitro (Supplementary Fig. 5). Moreover, phyB and SPA1 mutually enhanced phosphorylation of PIF1 in vitro (Fig. 5). Thus, it appears that multiple kinases are

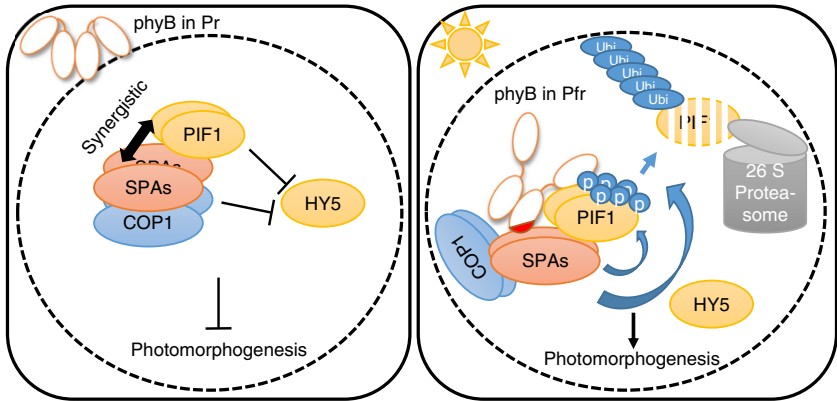

**Fig. 9** Model showing phyB-SPA1-COP1-PIF1 relationships in dark and light conditions. In darkness, inactive Pr form of phyB is present in the cytoplasm, while COP1-SPA complex together with PIF1 as a co-factor induces degradation of positively acting transcription factors (HY5/HFR1 and possibly others). However, upon light exposure, activated Pfr form of phyB translocates into the nucleus and interacts with PIF1 as well as SPA1 to trigger rapid phosphorylation of PIF1. phyB interacts with PIFs mostly through its N-terminal domain, while its extreme C-terminal 80 amino acids (shown as a red patch) are necessary for SPA1 interaction. By stabilizing the phyB-SPA1-PIF1 tripartite complex, phyB can initiate the light-induced phosphorylation of PIF1 by SPA1 kinase. Phosphorylated PIF1 is then recognized by the CUL4$^{COP1-SPA}$ E3 ubiquitin ligase for rapid poly-ubiquitination and subsequent degradation through the 26S proteasome. Degradation of PIFs and stabilization of HY5 result in promotion of photomorphogenesis

necessary for sequential and possibly cooperative phosphorylation of PIF1. Phenotypic analyses showed that only *spaQ* exhibits robustly delayed germination phenotype and an absence of the light-induced PIF1 phosphorylation in vivo. Therefore, SPAs are necessary for phyB-mediated signal transduction to regulate PIF1 abundance (Fig. 9).

Because SPAs are crucial for the COP1-SPA E3 ubiquitin ligase complex, this study identified a novel mechanism by which a cognate kinase-E3 ligase complex selectively interacts with its substrate and triggers rapid degradation in a light-inducible manner. This unique kinase-E3 ligase complex may account for the rapid degradation speed observed under red light for PIF1[38]. Finally, our findings suggest that other known COP1-SPA targets might also be phosphorylated by SPAs, given that the WD40 repeat domains present in COP1 and SPA proteins prefer phosphorylated substrates for binding[67,68]. Future experiments are necessary to address whether other COP1-SPA substrates are also phosphorylated by SPA kinase, and if the cognate kinase-E3 ubiquitin ligase model presented here may also be applicable to other E3 ligase complexes in plants and animals.

## Methods

**Plant materials and growth conditions**. Wild type Col-0, various mutants, and transgenic plants in the Col-0 background were used in this study unless otherwise indicated. Plants were grown in soil under 24-h light at 22 ± 0.5 °C.

**Construction of vectors and generation of transgenic plants**. *TAP-PIF1*, *TAP-SPA1*, *PIF1-HA*, *TAP-SPA1/PIF1-HA*, and *TAP-PIF1/spaQ* transgenic plants were reported previously[13,22,38,41,42]. To generate phyB-GFP FL and phyB-GFP ΔC, PHYB F KpnI-PHYB R SmaI, PHYB-GFP ΔC R SmaI primers were used to amplify PHYB coding sequence. These were first subcloned into pEZS-NL vector. The expression cassettes from pEZS-NL-PHYB(ΔC) were cut out with NotI and cloned into pART27 binary vector with NotI restriction enzyme. Then pART27-PHYB(ΔC)-GFP were transformed into *phyA-211phyB-9* double mutant and transformants were selected in the presence of Kanamycin antibiotics. For SPA1, full-length *SPA1-6his* coding sequence was amplified by PCR using the primers SPA1 F BamHI -His R XhoI and cloned into pCAMBIA C-luc vector. The mutant version of SPA1, mSPA1 was generated using Quickchage II site-directed mutagenesis kit (Agilent, Catalog #200523) in pBSK-SPA1. Both wild type and the mutant SPA1 were transformed into GV3101 and then transformed into *spa123* mutant. Obtained SPA1 transgenic plants in *spa123* were further crossed with *spaQ* mutant to generate SPA1/*spaQ* and mSPA1/*spaQ* transgenic plants.

**Protein purification from *E. coli***. SPA1 kinase domain was subcloned into pASK75 vector with SPA1 F BamHI- SPA1 kin R SalI primers. SPA1 kinase domain was then PCR amplified with SPA1 F BamHI-Strep R NotI primers and

cloned into pGEX4T-1. For generation of mSPA1, pASK75-SPA1 kinase domain clone was mutagenized using Quickchange II site-directed mutagenesis kit (Agilent, Catalog #200523). Each plasmid was transformed into BL21(DE3) cells. Protein expression was induced under 16 °C for 12 h with 0.1 mM IPTG. Collected cells were sonicated in binding buffer (100 mM Tris, pH 7.5, 150 mM NaCl, 0.2% Tergitol NP-40, 0.1 mM EDTA, 1· Protease inhibitor cocktail, and 1 mM PMSF) and purified using Strep-Tactin Sepharose column (IBA, Cat# 2-1209-001). Columns were washed three times thoroughly by applying 5 mL of the binding buffer. Proteins were eluted with the elution buffer (Tris 7.5 100 mM, EDTA 1 mM, 10% glycerol, 10 mM d-desthiobiotin) into separate fractions. The eluted proteins were analyzed on a SDS-PAGE gel and used for kinase assays. pGEX-strep clone was previously reported[50]. GST-PIF1-strep was purified using the same method as described in this section. For MBP-SPA1, pMAL p2x SPA1 construct was used as described previously[41].

**SPA1 and phytochrome purification from *Pichia pastoris***. The full-length SPA1 coding sequence was PCR-amplified with SPA1 F BamHI-SPA1 R SalI primers and subcloned into pASK75 vector. Then, SPA1-strep coding sequence was PCR-amplified with SPA1 F SnaBI- strep R NotI primers and cloned into pPIC3.5 K vector. Flag sequences (3·) were further cloned into the SnaBI site of the pPIC3.5 K SPA1-strep clone using Flag F-Flag R primers. Approximately 30 μg of SPA1-strep pPIC3.5 K plasmid DNA was linearized using PmeI restriction enzyme. *Pichia pastoris* strain GS115 was grown in YPD media and the electro-competent cell was prepared according to the manufacturer's instructions (Life Technology, Pichia Expression Kit, Cat.# K1710-01). Electroporation was performed using a general setup provided by electroporator (Bio-rad) for *Pichia pastoris*. Transformed colonies were selected on the MD media plates (1· YNB, 1· biotin, 2% Dextrose, 1.8% agar). Plates were incubated at 30 °C for two days to obtain ~5000 colonies. Multiple genomic integrations of the plasmids were further confirmed by antibiotic resistance at 0.5, 1, 1.5, 2, 2.5, 3, 3.5, and 4 μM of G418 Geneticin (Sigma, Cat.# 11811023) containing YPD media for five days of incubation. The selected colonies were tested their Mut$^s$ phenotype on solid MM media (1· YNB, 1· biotin, 5% methanol, 1.8% agar) and used for protein purification. For induction of the protein expression, the colony was initially inoculated in 10 mL of liquid MM media for 24 h, and then cultured in 200 mL of liquid MGY (1x YNB, 1x biotin, 2% glycerol) media for an additional 24 h. The cells were collected and MGY media was completely removed. Then, the collected cells were resuspended and diluted into 1 L of liquid MM media (1x YNB, 1x biotin, 5% methanol) to induce protein expression for 24 h at 30 °C.

The collected induced cells were resuspended in the lysis buffer (100 mM Tris pH 7.5, 150 mM NaCl, 0.2% Tergitol NP-40, 0.1 mM EDTA, 1x Protease inhibitor cocktail, 1 mM PMSF) and ground in liquid nitrogen using physical force with a homogenizer (Nohonseiki Kaisha LTD Japan, model AM-3). The whole cell extracts were filtered through a 0.45 μm filter and ran through the Strep-Tactin Sepharose column (IBA Lifesciences, Cat.# 2-1202-001) for purification. The column was washed 5 times thoroughly before eluting with the elution buffer (100 mM Tris pH 7.5, 1 mM EDTA, 10% glycerol, 10 mM d-desthiobiotin). Purified protein was analyzed on a SDS-PAGE gel.

Arabidopsis phytochrome B was purified from *Pichia pastoris* as described previously[69]. In brief, *PHYB* expressing *Pichia pastoris* GS115 strain was grown in 200 mL MGY media for 24 h before transferring into 1 L MM media to induce protein expression. The cells were harvested and homogenized. Total proteins were

precipitated by ammonium sulfate salting-out (0.23 g/mL) methods. The pellet was resuspended and assembled with a chromophore (10 μM), phycocyanobilin (Frontier Scientific, Cat.# P14137). Strep-Tactin Sepharose column was used to purify phyB holoprotein. The purified protein was analyzed on a SDS-PAGE gel.

**Phytochrome purification from *Saccharomyces cerevisiae*.** We obtained *Saccharomyces* strain RKY1293 from Dr. Arlen Johnson at UT. The full-length *PHYB* coding sequence was first subcloned into pEZS-NL using phyB F KpnI- phyB R SmaI primers. By using phyB F KpnI- GFP R XbaI primers, PHYB-GFP was PCR-amplified and cloned into pYES2 (Invitrogen, Carlsbad, USA) vector. The pYES2-PHYB-GFP was transformed into yeast strain RKY1293 using EZ transformation kit and transformants were selected on –URA dropout media. The selected colonies were grown in the 50 mL of liquid –URA dropout media with 2% raffinose as a carbon source. By adding 2% of galactose, phyB expression was induced for 12 h. The induced cells were collected and resuspended in lysis buffer (100 mM Tris pH 7.5, 150 mM NaCl, 0.1% Tergitol NP-40, 1 mM EDTA, 1x Protease inhibitor cocktail, 1 mM PMSF), and then frozen in liquid nitrogen to be ground thoroughly using a pestle and mortar. The fine powder was thawed on the mortar and collected in the test tubes. Tubes were centrifuged at a maximum speed 20200 × g for 20 min at 4 °C. For purification of the apoprotein, one microgram of anti-GFP (Abcam # ab6556) antibody was preincubated with 20 μL of protein A Dynabeads for 15 min. The antibody-bound Dynabeads were washed with the lysis buffer and incubated with the supernatant. After two hours of incubation, the Dynabeads were collected and washed three times with the lysis buffer using a magnet rack. The immuno-precipitated PHYB containing Dynabeads were resuspended in 100 μL of the lysis buffer supplemented with 10 ⌈M phycocyanobilin (PCB) and incubated in the dark for 1 h. The Dynabeads were collected and washed four times thoroughly with the lysis buffer and used for other assays.

**In vitro kinase assay.** For SPA1 kinase assay, about 500 ng of SPA1 and 1 μg of GST-PIF1 were used. For *Pichia* purified phyB, about 1 μg of holoprotein was used for the kinase assay. For *Saccharomyces* purified phyA and phyB, about 500 ng of holoproteins were used. All kinase assays were performed in kinase buffer (50 mM Tris, pH 7.5, 4 mM β-mercaptoethanol, 1 mM EDTA, 10 mM MgCl$_2$) except the phyB + SPA1 light dependent kinase assay, which contains 100 mM NaCl. $^{32}$P radio-labeled gamma-ATP (Perkin Elmer Cat# BLU502A) was added to the reaction and incubated at 28 °C for 1 h unless otherwise indicated. SDS sample buffer (6x) was added to stop the reaction and the boiled proteins were separated on a SDS-PAGE gel. The gels were dried and exposed to a phosphor screen and then scanned with Typhoon FLA 9500 (GE healthcare, Chicago, USA).

**Acid and base sensitivity assay.** The kinase assay was performed as described in the previous section. The total reaction containing SPA1 and PIF1 was boiled and divided into three lanes in a SDS-PAGE gel. The proteins were blotted onto PVDF membrane and cut into three pieces. Each piece was treated with neutral (Tris, pH 7.5), basic (3 M KOH) and acid (1 M HCl) solution as described previously[49]. The membrane was then exposed to a phosphor screen and scanned by Typhoon FLA 9500 (GE healthcare, Chicago, USA).

**PIF1 mobility shift assay.** To observe TAP-PIF1 mobility shift in Col-0 and *spaQ*, total protein was separated in a 10·10.5 cm 6.5% SDS-PAGE. For phostag gel analysis, each sample was ground in the buffer (50 mM Tris, pH 7.5, 150 mM NaCl, 1 mM EDTA, pH 8.0, 0.1% Tween20, 1 × protease inhibitor cocktail [Sigma-Aldrich, Cat# P9599], 1 mM PMSF, 20 μM bortezomib, 25 mM β-glycerophosphate, 10 mM NaF, and 2 mM Na orthovanadate) and immunoprecipitated using anti-myc (Sigma, Cat.# C3956) antibody. The immunoprecipitated TAP-PIF1 was then separated on a 6.5% SDS-PAGE gel containing 15-30 μM phostag acrylamide (Wako Japan, Cat# AAL-107). The phostag gel was prepared according to the manufacturer's instructions.

**Yeast two-hybrid Yeast two-hybrid assay.** LexA-phyB CT indicates the C-terminal domain of PHYB (aa 625-1172) and LexA-phyB CT ΔC indicates the C-terminal half of PHYB without the last 80 amino acids (aa 625-1092). For the bait proteins, phyB CT and phyB CT ΔC PCR products were amplified using PHYB-F-BamHI/PHYB-R-SalI and PHYB-F-BamHI /PHYB CT ΔC-R-SalI primers, respectively. PCR products were then cloned into pEG202 vector with LexA fusion. For the prey proteins, the full-length and various deletion fragments of AD-SPA1 fusion constructs were described previously[41]. All the constructed were confirmed by enzyme digestion and sequencing.

Plasmids containing bait and prey constructs were transformed into EGY48 yeast strain and selected on -His, -Ura, -Trp dropout medium at 30 °C for 3 days. Yeast colonies grown on this minimal medium were cultured overnight in liquid media supplemented with 2% (w/v) glucose. Overnight cultures (1 mL) were then transferred to 4 mL liquid medium supplemented with 2% (w/v) galactose and 1% (w/v) raffinose to induce the expression of proteins. β-galactosidase assays were performed according to the Matchmaker Two-Hybrid System, (Takara Biosciences). Three independent replicates were performed and the bar graphs represent the β-galactosidase activity normalized by activation domain (AD) vector control.

The light-dependent Yeast two-hybrid assays were performed as described[26]. The AD-PIF3, AD-SPA1, and D153-phyB NT-BD constructs have been described previously[41,70]. Briefly, AD and BD fusion combination plasmids were transformed into AH109 yeast cells and selected on drop out media without Leu and Trp. Individual colonies were cultured in the same liquid media and 10 μL of each culture was spotted on a plate containing 5 mM 3-AT and 20 μM phycocyanobilin (PCB), but lacking Leu, Trp and His. Colonies were grown at 30 °C for 2 days and then photographed. A control plate without Leu and Trp was grown to show normal growth.

**In vivo co-immunoprecipitation assay.** For in vivo co-immunoprecipitation assays, ~400 4-day-old dark-grown seedlings were collected and ground in liquid nitrogen. Total proteins were solubilized in the IP buffer (50 mM Tris, pH 7.5, 150 mM NaCl, 1 mM EDTA, pH 8.0, 0.1% Tween-20, 0.25 mM DTT, 1 × protease inhibitor cocktail [Sigma-Aldrich Co., St. Louis, MO, Cat# P9599], 1 mM PMSF, 20 μM bortezomib, 25 mM β-glycerophosphate, 10 mM NaF, and 2 mM Na orthovanadate), cleared by centrifugation at 20200 xg for 10 mins. Anti-HA (Abcam, Cat.# ab9110), and anti-GFP (Abcam, Cat.# ab6556) were bound to dynabeads before adding into total cell extractions. After 2 h of incubation at 4 °C in the dark, the dynabeads (20 μl/μg antibody; Life Technologies, Carlsbad, USA) were collected and washed using IP buffer on magnet rack. The beads were then boiled with 6X SDS sample buffer, ran on a 6.5% SDS-PAGE gel, and blotted with corresponding antibodies.

**In vitro/semi-in vitro pull-down assay.** For in vitro pull-down assay, GST-PIF1, GST, SPA1-strep, and mSPA1-strep proteins were purified as described above. Total 2 μg of GST and GST PIF1 proteins were bound to 25 μl of GST agarose beads (Pierce, Waltham, USA. Cat.# 20211). One microgram of SPA1 and mSPA1 was added to each sample with 500 μL binding buffer (50 mM Tris, pH 7.5, 150 mM NaCl, 1 mM EDTA, 0.1% Tergitol NP-40, 0.5 mM DTT, 1 × protease inhibitor cocktail, 1 mM PMSF). After 2 h of incubation with gentle rotation at 4 °C, beads were collected with centrifugation at 1800 × g. The collected beads were thoroughly washed five (x 5 mins) with 1 ml of the binding buffer by gentle rotation. Beads were then boiled with 2x SDS sample buffer and separated on an SDS-PAGE gel to visualize protein with Coomassie staining.

For semi-in vitro immunoprecipitation assay, phyB-GFP FL and phyB-GFP ΔC seeds were grown in the dark for 4 days before grinding in liquid nitrogen. Total protein was solubilized in IP buffer (50 mM Tris, pH 7.5, 150 mM NaCl, 1 mM EDTA, 0.1% Tween20, 0.25 mM DTT, 1 × protease inhibitor cocktail [Sigma-Aldrich Co., St. Louis, MO, Cat# P9599], 1 mM PMSF). The extracts were cleared by centrifugation at 20200 × g for 10 min. The supernatant was incubated with anti-GFP (Abcam, Cat.# ab6556) antibody-bound dynabeads (20 μL/μg antibody) for one hour in the dark. Immunoprecipitated phyB-GFP proteins were washed three times with IP buffer. Whole cell extract from IPTG induced *E. coli* harboring pMal p2x SPA1 plasmids were added into immunoprecipitated beads and each tube was treated either red (10 μmolm$^{-2}$ s$^{-1}$) or far-red light (15 μmole/s/m$^2$) for 5 mins. After an additional two hours incubation at 4 °C, beads were washed four times, five mins each with extraction buffer (100 mM phosphate buffer pH 7.8, 150 mM NaCl, 0.5 mM DTT, 0.2% Nondidet P-40, 1x protease inhibitor cocktail, 1 mM PMSF). Beads were then boiled with 2X SDS sample buffer and separated on a SDS-PAGE gel. Anti-MBP (NEB, Cat.# E8032S) antibody was used to detect co-immunoprecipitated SPA1-MBP protein.

For the tripartite complex pull-down assay, 2 μg of purified GST-PIF1-strep were bound to GST agarose beads. Twenty μL of the GST-bound beads were resuspended in the binding buffer (50 mM Tris, pH 7.5, 150 mM NaCl, 1 mM EDTA, 0.1% Tween 20, 0.25 mM DTT, 1 × protease inhibitor cocktail,1 mM PMSF). SPA1-MBP protein expression was induced in the BL21 cell as described in the previous section. The induced cell pellets were resuspended and sonicated in the lysis buffer (50 mM Tris, pH 7.5, 150 mM NaCl, 1 mM EDTA, 0.2% Tween 20, 0.25 mM DTT, 1 × protease inhibitor cocktail,1 mM PMSF), cleared by centrifugation at 20200 × g for 30 mins. The supernatant was then added to prepared GST-PIF1 beads and incubated with or without increasing amount of purified phyB (1–2 μg) from *Pichia pastoris*. After the pull-down, the beads were washed three times, five min each with the binding buffer. Co-precipitated MBP-SPA1, phyB and GST-PIF1 was immunoblotted with anti-MBP (NEB, Cat.# E8032S), anti-strep (Sigma, Cat.# GT661), and anti-GST HRP (GE healthcare, Cat. # RPN1236) antibodies.

**In vitro phytochrome dimerization assay.** Total 5 μg of *Pichia pastoris* purified phyB-GFP FL and ΔC holoproteins were mixed with 2X loading buffer (62.5 mM Tris-HCl, pH 6.8, 25% Glycerol, 1% Bromophenol Blue). Native 6% PAGE gel was prepared and ran in 4 °C, 100 v for 3 hours. The native PAGE gel was stained with Coomassie brilliant blue staining solution and the proteins were visualized.

**PIF1 sequestration assay using chromatin immunoprecipitation.** Chromatin immunoprecipitation (ChIP) was performed as previously described[71]. In brief, *phyB-GFP FL/TAP-PIF1* and *phyB-GFP ΔC/TAP-PIF1* were plated in MS media and grown for 4 days in the dark. Final 10 μM bortezomib was pretreated in the dark for 4 h before the red light irradiation to block 26 S proteasome-mediated

degradation of the TAP-PIF1. A pulse of red light (300 μmol s$^{-2}$) treated for the red samples and whole seedlings were subjected to cross-linking in 1% formaldehyde buffer for 15 min. The crosslinking was quenched by adding 2 M glycine and incubated additional 5 min. The seedlings were collected and ground in liquid nitrogen, resuspended in the nuclear isolation buffer (0.25 M sucrose, 15 mM PIPES, 5 mM MgCl2, 60 mM KCl, 15 mM NaCl, 1% Triton X-100, 2 mM PMSF, 1X protease inhibitor cocktail). Cleared by 15 min centrifugation at 20200 × g at 4 °C, the pellets were resuspended in 1 mL lysis buffer (50 mM HEPES pH7.5, 150 mM NaCl, 1 mM EDTA, 1% Triton X-100, 0.1% SDS, 2 mM PMSF, 1X protease inhibitor cocktail). Each sample was sonicated at 10% amplitude with Branson Digital Sonifier (Model# PC BD 400 W 20 kHz) sonication machine (10 seconds each for ten times), total 100 seconds sonication. Samples were centrifuged at 20,200 × g for 20 min. Ten percent of the supernatant was saved for input DNA isolation. Anti-myc (Sigma, Cat.# C3956) antibody-bound Dyanbeads were then added to the supernatant for PIF1 chromatin immunoprecipitation overnight. Bound ChIP samples were washed sequentially with low salt, high salt, LiCl, and TE wash buffer for 5 mins each. Eluted chromatin were then subjected to reverse crosslinking by overnight incubation in the presence of NaCl. After proteinase K treatment for an hour, DNA was isolated by DNA purification kit (Qiagen, Cat.# 28106). Enrichment was measured by quantitative PCR using primers corresponding to the G-box region on *PIL1* promoter and *PIL1* coding region were compared as a negative control.

**Protein extraction and western blot analyses**. Seeds were surface sterilized and grown in the dark for four-days otherwise indicated. For total protein extraction, about 100 seeds were plated. The tissue was collected and ground in 100 μL denaturing extraction buffer [100 mM Tris pH 7.5, 1 mM EDTA, 8 M Urea, 1x protease inhibitor cocktail (Sigma-Aldrich Co., St. Louis, MO, cat# 59), 2 mM PMSF, + optional 25 mM β-GP, 10 mM NaF and 2 mM Na-orthovanadate for PIF1 included samples] and cleared by centrifugation at 20,200 × g for 10 min at 4 °C. Samples were boiled for 5 min with 1X SDS sample buffer and separated on a 6.5% SDS-PAGE gel, blotted onto PVDF membranes, and probed with corresponding antibodies. Antibodies used in these studies are anti-PIF1[38], anti-myc (Sigma, Cat.# C3956 for immunoprecipitation, EMD Millipore, Cat#OP10-200UG for Western blot), anti-GFP (Abcam, Cat.# ab6556 for immunoprecipitation), anti-GFP (Santa Cruz. Cat.# sc-390394 for Western blot) anti-HA (Covance Cat.# MMS-101P for Western blot), anti-MBP (NEB, Cat.# E8032S), and anti-RPT5 antibodies (Enzo Life Sciences, Cat.# BML-PW8770-0025). Secondary HRP bound antibodies were visualized with Super Signal West Femto Chemiluminescent substrate (Pierce Biotechnology Inc., Waltham, USA.), and developed with an X-ray film. The PIF1 amount was quantified with image J from three independent blots and normalized by its corresponding RPT5 levels.

**Light-induced seed germination assay and measurement of hypocotyl lengths**. For measurement of hypocotyl lengths, the same batch of seeds were harvested and dried in the presence of desiccation rocks at room temperature for 1 month. Photographs of 4-day-old dark- and light-grown seedlings were taken and at least 20 seedlings were measured using the ImageJ software (http://rsb.info.nih.gov/ij/).

Seeds were harvested and dried at room temperature in the presence of desiccation rocks for one month. Light-induced germination assay was performed as described previously[35]. In brief, surface sterilized seeds were imbibed on the MS media for one hour. Plates were initially treated with far-red light (10 μmol/s/m$^2$) for 5 mins and then irradiated with red light with the amount of light as indicated in the figure. The plates were wrapped with aluminum foil twice and incubated at 22 °C for four days before counting germination frequency.

**RNAseq analysis**. Total six genotypes; Col-0, *cop1-4*, *spaQ*, *phyAphyB/phyB-GFP FL*, and *phyAphyB/phyB-GFP ΔC* plants were used for mRNA sequence analysis. The seeds were surface sterilized and plated on the MS media without sucrose, cold stratified for three days before treated for 3 hours of white light and 21 h of dark incubation to induce germination. After the initial 24 h, seeds were further treated with far-red light (10 μmolm$^{-2}$ s$^{-1}$) to inactivate phytochrome activity. After an additional two days in the dark, seedlings were either collected (dark sample) or treated with 1 h of continuous red light (red sample) (3.5 μmolm$^{-2}$ s$^{-1}$). Total RNA was isolated using the plant total RNA kit (Sigma, Cat.# STRN250). NEB-Next® Ultra™ DNA Library Prep Kit for Illumina® (NEB Cat.# E7370S) was used to generate mRNA sequencing library according to the manufacturer's instructions. The constructed library was quality checked by Agilent 2100 Bioanalyzer (Waldbroon, Germany) and sequenced on Illumina NextSeq 500 platform. Raw data and processed data were deposited into the Gene Expression Omnibus database (accession number GSE117114).

For the RNA-seq data analysis, the quality of raw reads (FASTQ) was initially checked using FastQC before aligned to the Arabidopsis genome using TopHat[72] calling Bowtie2[73]. Arabidopsis genome was acquired from TAIR10 (https://www.arabidopsis.org/).

HTSeq-count v0.8.0 was used to generate counts of reads mapped to annotated Arabidopsis genes[74] (http://htseq.readthedocs.io/en/master/index.html). The data preprocessing, mapping and counting reads were performed using TACC Stampede2 supercomputer.

Differentially expressed genes in each mutant compared to wild-type (WT) were analyzed using DESeq2 package in R programming based on the negative binomial distribution[75]. The differential gene expression from DESeq2 was selected for significant changes with the |log2FC| > 2 for Col-0, *spaQ*, *cop1-4* and |log2FC| > 1.5 for *phyB-GFP FL*, *phyB-GFP ΔC* with adjusted *p*-value < 0.05, Wald test as part of the DESeq2 package.

Venny v2.1.0 and ComplexHeatmap[76] package in R were used to generate Venn diagrams and heat maps. Enrichment analysis (Gene Ontology) was implemented using the Database for Annotation, Visualization and Integrated Discovery (DAVID) v6.8. In addition, to obtain the number of read per kilo bases per million reads (RPKM), the aligned reads from TopHat were assembled and quantified using Cufflinks[72]. Differentially expressed genes between two conditions were analyzed by using Cuffdiff from Cufflinks. Significantly difference was selected with adjusted *p*-value < 0.05, Cuffdiff analysis.

**Reporting summary**. Further information on research design is available in the Nature Research Reporting Summary linked to this article.

## Data availability

RNA sequencing data were deposited into the Gene Expression Omnibus database (accession numbers GSE112662 and GSE117114). Arabidopsis mutants and transgenic lines, as well as plasmids and antibodies generated during the current study are available from the corresponding author upon reasonable request. The authors declare that the data supporting the findings of this study including the gels and immunoblots in Figs. 1–7 and Supplementary Figs. 1–10 are available within the paper and its supplementary information files.

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

## Acknowledgements

We thank Dr. Karen Browning and the members of the Huq laboratory for critical reading of the paper, Akira Nagatani for sharing seeds, Peter Quail for sharing anti-phyB antibody. This work was supported by grants from the National Institute of Health (NIH) (GM-114297) and National Science Foundation (MCB-1543813) to E.H., and Next-Generation BioGreen 21 Program from Rural Development Administration, Republic of Korea (grant no. PJ01325301) to J.K. The authors acknowledge the Texas Advanced Computing Center (TACC) at The University of Texas at Austin for providing High Performance Computing, visualization, and database resources that have contributed to the research results reported in this paper.

## Author contributions

I.P., F.C., V.P., L.Z. and E.H. designed experiments. J.K. contributed essential reagent. I.P., F.C., V.P. and L.Z. carried out experiments. I.P., F.C., V.P., L.Z., J.K. and E.H. analyzed data. I.P. and E.H. wrote the paper. I.P., F.C., V.P., L.Z., J.K. and E.H. edited the paper.

## Additional information

**Competing interests:** The authors declare no competing interests.

