## [Peer Review File · Nature Communications]

Reviewers' comments:

Reviewer #1 (Remarks to the Author):

SPA1 has been discovered as suppressor of a weak phyA mutant and has (together with its family members SPA2-4) been identified to be part of the COP1 E3 ligase complex. The protein features an N-terminal kinase-like domain and a C-terminal coiled-coil and WD-repeat domain. Up to now no kinase activity of the SPA proteins has been reported and deletion of this part of the protein appears not to interfere with most of its functions (Holtkotte et al. 2016). The manuscript presents proof for SPA1 kinase activity and relates this activity to PIF 1 phosphorylation and PIF1s degradation.

The finding that SPA1 acts as kinase and the assigned functional relevance to light signalling are indeed important, interesting and novel discoveries. Nevertheless, I am not completely convinced, that the discovered kinase activity is specific and really necessary for SPA1 function. The authors show convincing evidence for the trimolecular phyB, PIF1 and SPA1 complex and its importance for PIF1 degradation. Since phyB alone can phosphorylate PIF, the necessity of an additional SPA1 kinase activity on PIF1 appears dispensable if SPA1 provides a scaffold for the partners. This line of arguments is partly covered by the findings of Holtkotte et al. 2016. Their results are relevant to the data presented by the authors and should be reflected upon.

Further points:

-The authors do not show any alignment which could be helpful to evaluate the importance of the mutated arginine 517 and to identify other functional important motifs. This is especially evident in relation to the mouse kinase used to underline the importance of the salt bridge. This alignment would be interesting. The focus is only on SPA1 although the motifs appear conserved in all four SPA family members. If the kinase activity of the kinase-like SPA1 domain is detectable although the similarity to established plant (mouse) kinases is rather low, the other family members should also show kinase activity since they share similar conserved islands in this domain.

-The in vitro kinase assay with the kinase-like domain alone is redundant with the following later experiments where full length protein was used with similar results. It could be left out and only mentioned in the section where the full length data are presented.

-A rat myelin was as efficiently phosphorylated as PIF1. This can be interpreted as indicating that the kinase activity of SPA1 is unspecific. Does autophosphorylation of SPA1 changes its affinity to COP1?

-In the GST pull down experiment (Fig. S1f) it appears that there is the same amount of SPA in input and in the pull down. I therefore doubt that the pull down is 100% efficient.

-In Fig. 2 where phosphorylation is linked to degradation the native PIF is gone after 10 min but the TAP:PIF appears quite stable for up to 20 min. This is surprising since the total protein amount (and hence also the amount of expressed TAP:PIF) appears to be much lower than in the blot with the endogenous PIF1, judging from the amount of RPT5 in both blots.

-The authors make the point that the SPA1-dependent phosphorylation is crucial for PIF1 degradation. In the quadruple spa mutant no phosphorylation can be detected and hence much less degradation is shown in Fig. 2. However, looking at a similar analysis in Fig. S2 I still see a lot of ubiquitination in spaQ with just less protein present (loading control?).

-The authors show that phosphorylation is light induced, but mSPA appears to have a phenotype in darkness. This should be discussed.

-The yeast-two-hybrid assay appears to be missing in the materials and methods section; it would be of interest if holo- or apo-phyB was used.

-Relating to Fig. 4c the text states that the proteins failed to interact but a slight band is still visible in blot.

-The authors state that: "... the PIF1-SPA1 interaction was further enhanced when the phyB amount was increased to two-fold". This is not visible in Fig. 4e.

-In regard to the Matsushita constructs Fig. S4e shows stabilization of PIF for both NG and BG. In the text only NG is mentioned which is in line with the arguments of the authors, but not shown in the figure.

Some minor points:

- In Fig. 6 the germination frequency x100% is already 100% on y axis.
- Some typing mistakes, for instance page 6 "could" should be corrected.
- There is a reference to Fig. S7c on page 8 but it is not mentioned that the SPA and mSPA were expressed as LUC-fusions.

Reviewer #2 (Remarks to the Author):

SPA1 has been known to act, together with COP1, as a master negative regulator in light signaling in plants. Although SPA1 protein has a serine/threonine kinase domain at its N-terminus, the kinase activity of SPA1 or its biological relevance has yet to be demonstrated. Here, in this work, Paik et al. showed that SPA1 acts as a serine/threonine kinase and directly phosphorylates PIF1, a major negatively acting component in the phytochrome-mediated light signaling pathway. The authors also showed that SPA1 family members are essential for the light-induced PIF1 phosphorylation and degradation *in vivo*. Moreover, they provided several biochemical data to claim that phytochrome interacts with SPA1 to enhance PIF1 phosphorylation by SPA1. Although it is an important advancement in this scientific field to show that SPA1 acts as a kinase to phosphorylate PIF1 in light signaling, many of the main claims in this study lack enough evidence and the interpretation of the data is somewhat biased. Detailed comments are shown below:

1) Although the authors showed that SPA1 directly interacts with and phosphorylates PIF1 *in vitro*, enough evidence is not provided to show that SPA1 directly phosphorylates PIF1 *in vivo*. More importantly, biological significance of PIF1 phosphorylation by SPA1 has not been directly investigated at all in this study. In order to clarify these points, SPA-dependent phosphorylation sites within PIF1 polypeptide should be analyzed and compared *in vitro* and *in vivo*. Then biological activity of PIF1 protein with these phosphorylation sites mutated should be examined in transgenic plants.

2) Since the authors' conclusion that phyB interacts directly with SPA1 through its C-terminus is based only on the yeast two-hybrid results in Figure 4a, several more lines of evidence are needed to confirm the direct interaction between phyB and SPA1. Moreover, although they claim that "the C-terminal 80 amino acids of phyB are indispensable for the light-induced SPA1 interaction" (p.9, l.17), Figures 4c and 4d clearly show that phyB(1-1092)-GFP is still capable of interacting with SPA1 in a light-dependent manner albeit weakly, which suggests that the N-terminal domain of phyB is sufficient for light-dependent interaction with SPA1. This is reminiscent of the case of phyB-PIF3 interaction in that the N-terminal domain of phyB is sufficient for light-dependent interaction with PIF3 while the C-terminal domain constitutively binds to PIF3, and that both the N- and C-terminal domains are required for stronger interaction with PIF3. Therefore, after confirming that the full-length phyB interacts with SPA1 in a light-dependent manner in the yeast two-hybrid assays supplemented with an appropriate chromophore such as phycocyanobilin, it has to be determined in the same assay whether the N-terminal domain of phyB can light-dependently bind to SPA1 or not. Moreover, in order to further confirm this point, the same *in vivo* co-immunoprecipitation assay and semi-*in vitro* pull-down assay as Figure 4c and 4d should be performed with the NG (phyB N terminal half-GFP-GUS-NLS) transgenic line, which completely lacks the C-terminal domain but retains full signaling activity.

3) One of the most important functions of the C-terminal domain of phyB is nuclear localization activity. Therefore, it has to be examined if phyB(1-1092)-GFP and phyB(1-991)-GFP show normal nuclear accumulation in response to red light in transgenic plants. Actually, it seems to the reviewer that all of the defects observed in the phyB(1-1092) and phyB(1-991) transgenic lines (Figures 5, 6, and S4) could be reasonably explained by reduced nuclear accumulation of phyB.

Minor comments:

4) In Figure 4e and 4f, full-length phyB holoprotein was used but the integrity of this protein was not shown. Similarly, although the authors claim that they established a method to purify functional phytochromes from yeast cells in Figure S3, it's not shown whether these phytochromes are really functional or not. Thus the integrity of these phytochrome proteins should be demonstrated, for example by showing that they can interact with PIF proteins in a red light-dependent manner.

5) The claim that "the PIF1-SPA1 interaction was strongly enhanced in the presence of the Pfr form of phyB (Fig. 4e) (p.9, l.24-25)" lacks any substantial evidence, because the apparent increase in the amount of co-pull-downed SPA1 protein might be simply due to direct interaction between phyB and SPA1.

6) In Figure 4f, it's not clear whether PIF1 phosphorylation by SPA1 and that by phyB have synergistic effects instead of additive effects to each other. Furthermore, again, the claim that "phyB enhances PIF1 phosphorylation by SPA1 in vitro (p.10, l.1)" lacks any evidence; for example, SPA1 could also enhance PIF1 phosphorylation by phyB.

7) In the experiment shown in Figure S4e, phyAphyB double mutants need to be analyzed as a negative control. Moreover, in this assay, the CG seedling also showed the same phenotype as the NG seedling, again indicating that not only the C-terminal domain but also the N-terminal domain of phyB are involved in rapid PIF1 degradation in response to red light, possibly through the interaction with SPA1.

Response to reviewer's comments

MS ID#: NCOMMS-18-33016

MS TITLE: A phyB-PIF1-SPA1 kinase regulatory complex promotes photomorphogenesis in Arabidopsis

AUTHORS: Inyup Paik, Fulu Chen, Vinh Ngoc Pham, Ling Zhu, Jeong-Il Kim and Enamul Huq

The specific changes and/or responses to the reviewers are listed below:

Reviewer #1 (Remarks to the Author):

SPA1 has been discovered as suppressor of a weak phyA mutant and has (together with its family members SPA2-4) been identified to be part of the COP1 E3 ligase complex. The protein features an N-terminal kinase-like domain and a C-terminal coiled-coil and WD-repeat domain. Up to now no kinase activity of the SPA proteins has been reported and deletion of this part of the protein appears not to interfere with most of its functions (Holtkotte et al. 2016). The manuscript presents proof for SPA1 kinase activity and relates this activity to PIF 1 phosphorylation and PIF1s degradation.

The finding that SPA1 acts as kinase and the assigned functional relevance to light signalling are indeed important, interesting and novel discoveries. Nevertheless, I am not completely convinced, that the discovered kinase activity is specific and really necessary for SPA1 function. The authors show convincing evidence for the trimolecular phyB, PIF1 and SPA1 complex and its importance for PIF1 degradation. Since phyB alone can phosphorylate PIF, the necessity of an additional SPA1 kinase activity on PIF1 appears dispensable if SPA1 provides a scaffold for the partners. This line of arguments is partly covered by the findings of Holtkotte et al. 2016. Their results are relevant to the data presented by the authors and should be reflected upon.

Author response: This is an important discussion in our field. In 2011 (Bu et al., JBC), we have shown that CK2 phosphorylates PIF1 and the CK2-mediated phosphorylation is necessary for PIF1

degradation. However, we concluded that CK2 is not the light regulated kinase. We hypothesized that multiple kinases might phosphorylate PIFs, which is necessary for the light-induced degradation of PIF1. In agreement with this hypothesis, PIF3 has been shown to be phosphorylated by PPK kinases. BIN2 has been shown to phosphorylate PIF4. Oat phyA has been shown to phosphorylate PIF3. Thus, there are multiple kinases for PIFs. In this manuscript, we show that SPA1 is a light-regulated kinase. This conclusion is based on multiple independent evidence: i) SPA1 can directly phosphorylate PIF1 *in vitro*, ii) overexpression of *SPA1* results in faster phosphorylation than control plants, iii) light-induced phosphorylation is eliminated in *spaQ* background under the conditions examined, iv) light-induced ubiquitination and degradation is reduced or absent in *spaQ* background, v) we show that the kinase domain of SPA1 behaves like a kinase as supported by ATP binding, ATP-dependent phosphorylation assays, time course, concentration-dependent phosphorylation of the substrate, and acid-base sensitivity assay, vi) finally, mutation in a critical residue in the kinase domain of SPA1 results in reduced phosphorylation *in vitro* and *in vivo*, and reduced degradation of PIF1 under light *in vivo*. These data convincingly show that SPA1 acts as a bona fide kinase for PIF1. We also hypothesize that there might be other kinases that are acting on PIF1 under longer light conditions as PIF1 is still degraded even in *spaQ* background (Zhu et al., 2015).

The hypocotyl data presented in our manuscript are consistent with Holtkotte et al 2016. Based on the same point mutation, these authors concluded that “the kinase domain has been evolutionarily conserved as it may provide structural information critical for SPA1 function in darkness”. These authors mostly focused on assays measuring hypocotyl lengths and leaf expansion, however, they did not examine the biochemical activity of these SPA1 mutants in terms of kinase assays and did not test seed germination for these lines. We are providing biochemical data suggesting that the kinase domain as well as the full-length SPA1 protein has kinase activity and this activity is necessary for SPA1-mediated regulation of seed germination through controlling PIF1 level. A new paragraph has been added to discuss our data in light of this paper.

Further points:

-The authors do not show any alignment which could be helpful to evaluate the importance of the

mutated arginine 517 and to identify other functional important motifs. This is especially evident in relation to the mouse kinase used to underline the importance of the salt bridge. This alignment would be interesting. The focus is only on SPA1 although the motifs appear conserved in all four SPA family members. If the kinase activity of the kinase-like SPA1 domain is detectable although the similarity to established plant (mouse) kinases is rather low, the other family members should also show kinase activity since they share similar conserved islands in this domain.

Author response: This is an excellent suggestion. We have now added the alignment of this region from all four SPA proteins from Arabidopsis as well as SPA1 homologs from other plants (new figure S1). This residue along with other residues in this region are identical among all the sequences examined, suggesting that this domain is evolutionarily conserved and might provide critical function for SPA proteins.

-The in vitro kinase assay with the kinase-like domain alone is redundant with the following later experiments where full length protein was used with similar results. It could be left out and only mentioned in the section where the full length data are presented.

Author response: These data are critical to establish that the kinase activity displayed by full-length SPA1 purified from a eukaryotic host is not due to a co-purifying kinase that may not be visible on gels. Thus, we are keeping these data as part of Fig. 1. The reviewer might be aware of this debate about co-purifying kinase activity for plant phytochromes for long time in our field.

-A rat myelin was as efficiently phosphorylated as PIF1. This can be interpreted as indicating that the kinase activity of SPA1 is unspecific. Does autophosphorylation of SPA1 changes its affinity to COP1?

Author response: The rat myelin basic protein (MBP) is a universal substrate for most eukaryotic kinases both in plants and animal system. It is usually used to show kinase activity for a new kinase like proteins. In fact, it has been used to show kinase activity for phytochromes as well (Shin et al., 2016, Nature Comm).

[redacted]

-In the GST pull down experiment (Fig. S1f) it appears that there is the same amount of SPA in input and in the pull down. I therefore doubt that the pull down is 100% efficient.

Author response: Thanks for pointing this out. The input is 5% of SPA1 used for pull-down, now indicated in figure and figure legend (New fig. S2).

-In Fig. 2 where phosphorylation is linked to degradation the native PIF is gone after 10 min but the TAP:PIF appears quite stable for up to 20 min. This is surprising since the total protein amount (and hence also the amount of expressed TAP:PIF) appears to be much lower than in the blot with the endogenous PIF1, judging from the amount of RPT5 in both blots.

Author response: The rate of degradation for native PIF1 and any fusion PIF1 (TAP-PIF1 and LUC-PIF1) are different (Shen et al., 2005, Shen et al., 2008). Native PIF1 degradation is much faster than TAP-PIF1, that's why the difference in band intensity at the same time point in these figures.

-The authors make the point that the SPA1-dependent phosphorylation is crucial for PIF1 degradation. In the quadruple spa mutant no phosphorylation can be detected and hence much less degradation is shown in Fig. 2. However, looking at a similar analysis in Fig. S2 I still see a lot of ubiquitination in spaQ with just less protein present (loading control?).

Author response: The ubiquitination assays is usually performed in the presence of proteasome inhibitor. Thus, it stabilizes small amount of ubiquitinated proteins. PIF1 is actually degraded even

in *spaQ* background under higher amount of light and longer time point than shown here (Zhu et al., 2015). Thus, SPAs are not the only kinases and E3 ligases acting on PIF1.

-The authors show that phosphorylation is light induced, but mSPA appears to have a phenotype in darkness. This should be discussed.

Author response: As shown in Holtkotte et al 2016, we also observed this phenotype. However, the biochemical mechanism of this activity is unknown. As this reviewer pointed out earlier, this mutation might affect dimerization with COP1. Alternatively, the kinase activity might be necessary for the degradation of other positively acting factors. We hope to answer these questions in future studies. We added a paragraph in discussion section about this topic.

-The yeast-two-hybrid assay appears to be missing in the materials and methods section; it would be of interest if holo- or apo-phyB was used.

Author response: It is added now, thanks for pointing out.

-Relating to Fig. 4c the text states that the proteins failed to interact but a slight band is still visible in blot.

Author response: We agree that SPA1 can still bind to phyB ΔC in *in vivo* assays. Our hypothesis is that phyB can hetero-dimerize with other phys, resulting in small amount of interaction. To rule out this possibility, we now have performed *in vitro* assays using purified phyB and SPA1 protein (Fig. 4e, f). The *in vitro* pull-down assay shows that phyB ΔC does not bind to SPA1. The quality of the phyB preparations is shown by *in vitro* light-dependent interaction with GST-PIF1 (Fig. 4f).

-The authors state that: "... the PIF1-SPA1 interaction was further enhanced when the phyB amount was increased to two-fold". This is not visible in Fig. 4e.

Author response: We have now repeated this assay and also quantified the interactions. The new data is presented (Fig. 5a, b). These data clearly show that phyB can enhance the interaction between PIF1 and SPA1.

-In regard to the Matsushita constructs Fig. S4e shows stabilization of PIF for both NG and BG. In the text only NG is mentioned which is in line with the arguments of the authors, but not shown in the figure.

Author response: We now present new data showing the stabilization of PIF1 in both *NG* and *CG* lines under light compared to *PBG* (new figure S5e).

Some minor points:

-In Fig. 6 the germination frequency x100% is already 100% on y axis.

Author response: This has been corrected.

-Some typing mistakes, for instance page 6 "could" should be corrected.

Author response: This has been corrected. Thanks for catching this mistake.

-There is a reference to Fig. S7c on page 8 but it is not mentioned that the SPA and mSPA were expressed as LUC-fusions.

Author response: This has been corrected.

Reviewer #2 (Remarks to the Author):

SPA1 has been known to act, together with COP1, as a master negative regulator in light signaling

in plants. Although SPA1 protein has a serine/threonine kinase domain at its N-terminus, the kinase activity of SPA1 or its biological relevance has yet to be demonstrated. Here, in this work, Paik et al. showed that SPA1 acts as a serine/threonine kinase and directly phosphorylates PIF1, a major negatively acting component in the phytochrome-mediated light signaling pathway. The authors also showed that SPA1 family members are essential for the light-induced PIF1 phosphorylation and degradation *in vivo*. Moreover, they provided several biochemical data to claim that phytochrome interacts with SPA1 to enhance PIF1 phosphorylation by SPA1. Although it is an important advancement in this scientific field to show that SPA1 acts as a kinase to phosphorylate PIF1 in light signaling, many of the main claims in this study lack enough evidence and the interpretation of the data is somewhat biased. Detailed comments are shown below:

1) Although the authors showed that SPA1 directly interacts with and phosphorylates PIF1 *in vitro*, enough evidence is not provided to show that SPA1 directly phosphorylates PIF1 *in vivo*. More importantly, biological significance of PIF1 phosphorylation by SPA1 has not been directly investigated at all in this study. In order to clarify these points, SPA-dependent phosphorylation sites within PIF1 polypeptide should be analyzed and compared *in vitro* and *in vivo*. Then biological activity of PIF1 protein with these phosphorylation sites mutated should be examined in transgenic plants.

Author response: This conclusion that SPA1 acts as a light-regulated kinase for PIF1 is based on multiple independent evidence: i) SPA1 can directly phosphorylate PIF1 *in vitro*, ii) overexpression of *SPA1* results in faster phosphorylation than control plants, iii) light-induced phosphorylation is eliminated in *spaQ* background under the conditions examined, iv) light-induced ubiquitination and degradation is reduced or absent in *spaQ* background, v) we show that the kinase domain of SPA1 behaves like a ser/thr kinase as supported by ATP binding, ATP-dependent phosphorylation assays, time course, concentration-dependent phosphorylation of the substrate, and acid-base sensitivity assay, vi) finally, mutation in a critical residue in the kinase domain of SPA1 results in reduced phosphorylation and degradation of PIF1 under light *in vivo*. These data convincingly show that SPA1 acts as a bona fide kinase for PIF1.

[redacted]

2) Since the authors' conclusion that phyB interacts directly with SPA1 through its C-terminus is based only on the yeast two-hybrid results in Figure 4a, several more lines of evidence are needed to confirm the direct interaction between phyB and SPA1. Moreover, although they claim that "the C-terminal 80 amino acids of phyB are indispensable for the light-induced SPA1 interaction" (p.9, l.17), Figures 4c and 4d clearly show that phyB (1-1092)-GFP is still capable of interacting with SPA1 in a light-dependent manner albeit weakly, which suggests that the N-terminal domain of phyB is sufficient for light-dependent interaction with SPA1. This is reminiscent of the case of phyB-PIF3 interaction in that the N-terminal domain of phyB is sufficient for light-dependent interaction with PIF3 while the C-terminal domain constitutively binds to PIF3, and that both the N- and C-terminal domains are required for stronger interaction with PIF3. Therefore, after confirming that the full-length phyB interacts with SPA1 in a light-dependent manner in the yeast two-hybrid assays supplemented with an appropriate chromophore such as phycocyanobillin, it has to be determined in the same assay whether the N-terminal domain of phyB can light-dependently bind to SPA1 or not. Moreover, in order to

further confirm this point, the same *in vivo* co-immunoprecipitation assay and semi-*in vitro* pull-down assay as Figure 4c and 4d should be performed with the NG (phyB N terminal half-GFP-GUS-NLS) transgenic line, which completely lacks the C-terminal domain but retains full signaling activity.

Author response: We agree with the reviewer that SPA1 can still weakly bind to phyB Δ C in *in vivo* assays (Fig. 4c, d). Although SPA1 binding to both N- and C-terminal halves of phyB similar to PIF3 is an excellent idea, our hypothesis is that phyB can hetero-dimerize with other phys in these *in vivo* and semi-*in vivo* co-IP assays, resulting in small amount of interaction. To rule out this possibility, we now have performed *in vitro* pull-down assays using purified phyB and SPA1 proteins from yeast (phyB) and bacteria (MBP-SPA1) (Fig. 4e, f). The *in vitro* pull-down assay shows that phyB Δ C does not bind to SPA1. These data are consistent with Zheng et al (2013) Plant Cell, where they showed the C-terminal 548 amino acids of phyB interacts with the coiled-oil domain of SPA1. The quality of the phyB preparation is also shown by *in vitro* light-dependent interaction with GST-PIF1 (Fig. 4f).

3) One of the most important functions of the C-terminal domain of phyB is nuclear localization activity. Therefore, it has to be examined if phyB (1-1092)-GFP and phyB (1-991)-GFP show normal nuclear accumulation in response to red light in transgenic plants. Actually, it seems to the reviewer that all of the defects observed in the phyB (1-1092) and phyB (1-991) transgenic lines (Figures 5, 6, and S4) could be reasonably explained by reduced nuclear accumulation of phyB.

Author response: This is an excellent suggestion. Although we did not have phyB (1-991)-GFP, we do have phyB-GFP FL and phyB-GFP (1-1092) which is phyB-GFP Δ C. We performed light-dependent nuclear localization of these proteins and observed that phyB-GFP FL can translocate into nucleus and form nuclear photobodies. phyB-GFP Δ C accumulated into the nucleus in response to light (Fig. 6e). However, phyB-GFP Δ C failed to form nuclear photobodies even after prolonged light conditions (16 hrs of red light) (Fig. 6e, S5). Moreover, we observed that phyB-GFP Δ C has a defect in nuclear transport compared to phyB-GFP FL (Fig. 6f). This domain is necessary for the light-dependent SPA1 interaction both *in vitro* and *in vivo*. SPA proteins have been shown to co-translocate with phyB into the nucleus¹. It is possible that this domain interacts

with SPA proteins as well as other factors, and these combined interactions are necessary for robust phyB nuclear transport and photobody formation. These data have been added in the result and discussion sections in the manuscript.

I agree with the reviewer that this causal relationship be a challenging issue to resolve. If this domain is necessary for interaction with SPA1 as well other factors, deletion of this domain in phyB results in defects in all of these interactions. In the same way, the coiled-coil domain of SPA1 interacts with phyB as well as COP1. If we delete CC domain in SPA1, it becomes non-functional due to structural issues as well as defects in these interactions. Based on current data we added the following paragraph in the discussion section:

“Although, these data highlight the importance of the C-terminal domain of phyB in regulation of PIF abundance, our data show that the last 80 amino acids of the C-terminal domain of phyB is necessary for the light-dependent SPA1 interaction both *in vitro* and *in vivo*. Moreover, this domain is necessary for phyB nuclear translocation and photobody formation in response to light (Fig. 6e). SPA proteins have been shown to co-translocate with phyB into the nucleus ¹. However, it is highly unlikely that SPA proteins are the only factors facilitating transport of phyB into nucleus, as PIFs have shown to promote nuclear translocation of phyB ². It is possible that this domain interacts with SPA proteins as well as other factors, and these combined interactions are required for phyB nuclear transport and photobody formation. Because phyB-GFP Δ C behaves normally in terms of chromophore binding (zinc blot),_dimerization, PIF1 interaction and sequestration, and complements the *phyB* phenotypes to a large extent, the defects observed in *phyB-GFP* Δ C seedlings might not be due to a non-functional protein, rather due in part to lack of phyB-SPA1 binding in addition to other potential unknown interactions necessary for robust phyB nuclear transport and photobody formation.”

Minor comments:

4) In Figure 4e and 4f, full-length phyB holoprotein was used but the integrity of this protein was not shown. Similarly, although the authors claim that they established a method to purify functional phytochromes from yeast cells in Figure S3, it's not shown whether these phytochromes

are really functional or not. Thus the integrity of these phytochrome proteins should be demonstrated, for example by showing that they can interact with PIF proteins in a red light-dependent manner.

Author response: The integrity of the phytochrome preparations is shown by *in vitro* light-dependent interaction with GST-PIF1 (new Fig. 4f, new S4b).

5) The claim that “the PIF1-SPA1 interaction was strongly enhanced in the presence of the Pfr form of phyB (Fig. 4e) (p.9, l.24-25)” lacks any substantial evidence, because the apparent increase in the amount of co-pull-downed SPA1 protein might be simply due to direct interaction between phyB and SPA1.

Author response: The assay is set up in a way that GST-PIF1 is the bait and MBP-SPA1 is the prey. We added two concentrations of phyB to examine if phyB can affect the interactions between PIF1-SPA1. We have now repeated this assay and also quantified the interactions. The new data is presented as Fig. 5a, b. These data clearly show that phyB can enhance the interaction between PIF1 and SPA1. It is possible that phyB is pull-downed because it interacts with SPA1. But we also know that phyB can interact with PIF1. Thus, it is forming a trimolecular complex. If we add increased amount of one component (in this case phyB), we observe even more of this trimolecular complex, suggesting that PIF1-SPA1 interaction is also enhanced in the presence of phyB. We now modified our claim to reflect this.

6) In Figure 4f, it's not clear whether PIF1 phosphorylation by SPA1 and that by phyB have synergistic effects instead of additive effects to each other. Furthermore, again, the claim that “phyB enhances PIF1 phosphorylation by SPA1 *in vitro* (p.10, l.1)” lacks any evidence; for example, SPA1 could also enhance PIF1 phosphorylation by phyB.

Author response: We modified the sentence to reflect that we observed increased phosphorylation of PIF1 in the presence of phyB and SPA1. This could be due to mutual enhancement of PIF1 phosphorylation by each other.

7) In the experiment shown in Figure S4e, phyAphyB double mutants need to be analyzed as a negative control. Moreover, in this assay, the CG seedling also showed the same phenotype as the NG seedling, again indicating that not only the C-terminal domain but also the N-terminal domain of phyB are involved in rapid PIF1 degradation in response to red light, possibly through the interaction with SPA1.

Author response: We now present new data adding *phyAphyB* mutants in this assay. We agree with the reviewer that PIF1 is still degraded in both *NG* and *CG* lines under light compared to *PBG* (new figure S5e). However, this degradation might be due to other factors (e.g., PPK/LRB-mediated degradation). This needs to be examined in future studies.

We hope that these modifications and our responses successfully address the reviewers' concerns.

References:

1. Zheng, X. *et al.* Arabidopsis Phytochrome B Promotes SPA1 Nuclear Accumulation to Repress Photomorphogenesis under Far-Red Light. *Plant Cell* **25**, 115-133 (2013).
2. Pfeiffer, A. *et al.* Interaction with plant transcription factors can mediate nuclear import of phytochrome B. *Proceedings of the National Academy of Sciences* **109**, 5892-5897 (2012).

Reviewers' comments:

Reviewer #1 (Remarks to the Author):

Paik et al. resubmission

The authors have addressed many of the points raised in the first review and added new experimental data. They present evidence that SPA1 can phosphorylate PIF1 which in turn leads to PIF1 degradation. This is also in line with the PIF1 stabilisation in the *spaQ* background. Although a kinase function of SPA1 is strongly supported by the data presented, the claim that light regulates this specific function has not become clearer in the new version of the manuscript. In Fig 2c WT SPA1 levels had no effect on (avowedly overexpressed) PIF1-HA and in Fig 5a no difference in PIF1 migration behaviour can be seen comparing D with ++R. There is also no difference between D and R visible in Fig. 5c. An alternative interpretation would be that SPA1 is light-dependently recruited into a complex with PIF1 and phyB but not necessarily by its kinase activity. I would therefore suggest that the light-dependent kinase arguments in the text be toned down and a scaffold explanation be presented as well. Since other factors appear also to phosphorylate PIF1, statements as in line 116 that SPA1 is necessary for PIF1 phosphorylation should be softened.

The authors also present data on the COP1-SPA1 interaction in a separate document (for reviewers only). The missing difference could be due to the missing phyB, which would require further *in vivo* experiments and therefore their reason for not including the preliminary result in the manuscript is acceptable.

Another unsolved issue is SPA1 function in darkness, which is apparently dependent on SPA1's kinase activity but somehow contradicts the light-induced kinase argument. The authors have included a new paragraph dealing with this issue in the discussion. Still, this remains an unsolved problem, weakening somewhat the role the authors confer to SPA1 in the manuscript.

Other minor issues:

Line 217-221: Fig 3 b and c are swapped, Fig 3b D (dark) should be indicated in figure or legend

The material and methods sections should be proofread by a native speaker. There are many (!) instances where the text is not correct up to the point of not making sense. For example: "...then red light was treated as indicated.../...for overnight.../...run for SDS gels.../... cleared by centrifuge..." etc.

Reviewer #2 (Remarks to the Author):

The authors have not fully addressed one of the reviewer's major concerns that not only the C-terminal half but also the N-terminal half of phyB may be sufficient for the interaction with SPA1. First of all, in Fig. 4a, it is not shown that the C-terminal half of phyB is required for the interaction with SPA1. In this figure, "PHYB FL" means the C-terminal half of phyB (aa 625-1172) but not the full-length phyB, which is quite confusing and even deceptive. Therefore, again, after confirming that the full-length phyB interacts with SPA1 in a light-dependent manner in the yeast two-hybrid assays supplemented with PCB, it has to be determined in the same assay whether the N-terminal half of phyB can light-dependently bind to SPA1 or not. Moreover, the reviewer still strongly believes that the same *in vivo* co-immunoprecipitation assay and semi-*in vitro* pull-down assay as Figure 4c and 4d should be performed with the NG (phyB N terminal half-GFP-GUS-NLS) transgenic line, which can directly test the authors' hypothesis that phyB(1-1092)-GFP heterodimerizes with other type II phytochromes to show light-dependent interaction with SPA1 in these

assays, since NG does not have any hetero-dimerizing activity.

The statement that “phyB enhances PIF1 phosphorylation by SPA1 in vitro” (p.10, l.263) is not based on any evidence, which should be rephrased.

Similarly, the conclusion that “phyB-GFP Δ C is specifically defective in SPA1 interaction and the phyB-SPA1 interaction is necessary for the light-induced degradation of PIF1 in Arabidopsis” (p.13, l.368-370 and p.10, l.267-268) should also be rephrased, because all of the defects observed in the phyB(1-1092) and phyB(1-991) transgenic lines (Figures 6, 7, and S6) can be explained not only by defects in SPA1 interaction but also by reduced nuclear accumulation of phyB.

In Fig. S6e, all of the four lines (phyAB/PBG, phyAB/NG, phyAB/CG, and phyAB) should be analyzed in the same blot.

Response to reviewer's comments

MS ID#: NCOMMS-18-33016A

MS TITLE: A phyB-PIF1-SPA1 kinase regulatory complex promotes photomorphogenesis in Arabidopsis

AUTHORS: Inyup Paik, Fulu Chen, Vinh Ngoc Pham, Ling Zhu, Jeong-Il Kim and Enamul Huq

The specific changes and/or responses to the reviewers are listed below:

Reviewer #1 (Remarks to the Author):

The authors have addressed many of the points raised in the first review and added new experimental data. They present evidence that SPA1 can phosphorylate PIF1 which in turn leads to PIF1 degradation. This is also in line with the PIF1 stabilisation in the *spaQ* background. Although a kinase function of SPA1 is strongly supported by the data presented, the claim that light regulates this specific function has not become clearer in the new version of the manuscript. In Fig 2c WT SPA1 levels had no effect on (avowedly overexpressed) PIF1-HA and in Fig 5a no difference in PIF1 migration behaviour can be seen comparing D with ++R. There is also no difference between D and R visible in Fig. 5c. An alternative interpretation would be that SPA1 is light-dependently recruited into a complex with PIF1 and phyB but not necessarily by its kinase activity. I would therefore suggest that the light-dependent kinase arguments in the text be toned down and a scaffold explanation be presented as well. Since other factors appear also to phosphorylate PIF1, statements as in line 116 that SPA1 is necessary for PIF1 phosphorylation should be softened.

Author response: We appreciate the reviewer's new concerns about Fig. 2c and Fig. 5a. In Fig. 2c, we are only examining the light-induced phosphorylation *in vivo* by an overexpressed *TAP-SPA1*. It was intentionally done in a way where wt SPA1 will have little or no effect whereas overexpression of *TAP-SPA1* will have an effect. This was done under very weak red light and also in the presence of a proteasome inhibitor to prevent degradation. For Fig. 5c, we have acknowledged the weakness in the manuscript that this assay did not show any band shift. There

was strong phosphorylation in the presence of both SPA1 and phyB, but may not be sufficient to display a band shift in this assay condition. One explanation could be due to multiple kinases acting on PIF1. We have shown that CK2 phosphorylates PIF1 and CK2-mediated phosphorylation is not light-dependent. Thus, phosphorylation by multiple kinases *in vivo* might be necessary to show the band shift that we do not see under our *in vitro* assay condition. This hypothesis is consistent with the lack of a band shift in *spaQ* background under *in vivo* conditions. We disfavor the scaffold hypothesis as we have shown the importance of the kinase activity in light-induced phosphorylation and degradation *in vivo* using a mSPA1 that showed reduced kinase activity *in vitro* (Fig. 1b, g) and *in vivo* (Fig. 3b).

The authors also present data on the COP1-SPA1 interaction in a separate document (for reviewers only). The missing difference could be due to the missing phyB, which would require further *in vivo* experiments and therefore their reason for not including the preliminary result in the manuscript is acceptable.

Author response: Thank you for understanding!

Another unsolved issue is SPA1 function in darkness, which is apparently dependent on SPA1's kinase activity but somehow contradicts the light-induced kinase argument. The authors have included a new paragraph dealing with this issue in the discussion. Still, this remains an unsolved problem, weakening somewhat the role the authors confer to SPA1 in the manuscript.

Author response: We agree that this is an important issue that requires further experiments. Dr. Hoecker lab has shown this phenotype first in 2016 and we are expecting that she will look into this issue more thoroughly. Our story is more focused on SPA1's role in light-regulated protein degradation, adding mechanistic details on the SPA1's role in darkness will extend the manuscript too much and also draws away attention from the main focus.

Other minor issues:

Line 217-221: Fig 3 b and c are swapped, Fig 3b D (dark) should be indicated in figure or legend.

Author response: This has been fixed—thanks for pointing out.

The material and methods sections should be proofread by a native speaker. There are many (!) instances where the text is not correct up to the point of not making sense. For example: "...then red light was treated as indicated.../...for overnight.../...run for SDS gels.../... cleared by centrifuge..." etc.

Author response: These have been fixed as well. Thank you for pointing out.

Reviewer #2 (Remarks to the Author):

The authors have not fully addressed one of the reviewer's major concerns that not only the C-terminal half but also the N-terminal half of phyB may be sufficient for the interaction with SPA1. First of all, in Fig. 4a, it is not shown that the C-terminal half of phyB is required for the interaction with SPA1. In this figure, "PHYB FL" means the C-terminal half of phyB (aa 625-1172) but not the full-length phyB, which is quite confusing and even deceptive. Therefore, again, after confirming that the full-length phyB interacts with SPA1 in a light-dependent manner in the yeast two-hybrid assays supplemented with PCB, it has to be determined in the same assay whether the N-terminal half of phyB can light-dependently bind to SPA1 or not. Moreover, the reviewer still strongly believes that the same *in vivo* co-immunoprecipitation assay and semi-*in vitro* pull-down assay as Figure 4c and 4d should be performed with the NG (phyB N terminal half-GFP-GUS-NLS) transgenic line, which can directly test the authors' hypothesis that phyB(1-1092)-GFP hetero-dimerizes with other type II phytochromes to show light-dependent interaction with SPA1 in these assays, since NG does not have any hetero-dimerizing activity.

Author response: This is an excellent suggestion. We have now performed semi *in vivo* co-immunoprecipitation assays using PBG and NG plants. Results show that PBG can interact with MBP-SPA1 in a light-stimulated manner. However, NG failed to interact with SPA1 under the same condition. Moreover, in yeast two-hybrid growth assays, SPA1 interacted with the C-terminal domain of phyB only, but not the N-terminal domain. However, PIF3 interacted with

the N-terminal half of phyB in a light-dependent manner under the same condition. PIF3 also interacted with the C-terminal domain of phyB as has been published previously by Quail laboratory. These data have now been added as Figure S4. Please note that the SPA1-phyB CT or PIF3-phyB CT interaction is light-independent. However, we placed these combinations under red light in the figure S4b, as we grew these along with NG under red light. We have not been able to show any interaction between SPA1 and full-length phyB for unknown reason after repeated effort. We will investigate this further in future.

We have also changed the description of the constructs from previous yeast two-hybrid assay (Fig. 4a), PHYB FL into PHYB CT and PHYB Δ C into PHYB CT Δ C. Sorry for the confusion, it was not meant to be deceptive as we clearly described the amino acid coordinates in figure legends.

The statement that “phyB enhances PIF1 phosphorylation by SPA1 in vitro” (p.10, 1.263) is not based on any evidence, which should be rephrased.

Author response: This statement has been rephrased.

Similarly, the conclusion that “phyB-GFP Δ C is specifically defective in SPA1 interaction and the phyB-SPA1 interaction is necessary for the light-induced degradation of PIF1 in Arabidopsis” (p.13, 1.368-370 and p.10, 1.267-268) should also be rephrased, because all of the defects observed in the phyB (1-1092) and phyB (1-991) transgenic lines (Figures 6, 7, and S6) can be explained not only by defects in SPA1 interaction but also by reduced nuclear accumulation of phyB.

Author response: These statements have been rephrased.

In Fig. S6e, all of the four lines (phyAB/PBG, phyAB/NG, phyAB/CG, and phyAB) should be analyzed in the same blot.

Author response: This has been done and the new data has been used to replace the old data in Fig. S6e. The results are similar to the last experiment where NG and CG show slight

degradation of PIF1 in response to light. We believe this is due to other pathways acting on PIF1 through phyAB.

Reviewers' comments:

Reviewer #1 (Remarks to the Author):

The manuscript has now been changed in some places to soften the previous hard claims of the authors. Nevertheless, several points still need attention and I do not agree on the authors disfavor for the scaffold hypothesis. I also think that the dark activity contradiction should be mentioned honestly –but agree with the authors that this is further project.

Further points (no specific order):

In the summary and as a heading the authors state

31 phyB interacts with SPA1 through its C-terminus and enhances the recruitment of PIF1 for phosphorylation

To my understanding SPA1 enhances the recruitment of PIF1 to phyB for phosphorylation?

108 "no PIF kinase known" contradicts the senior author's own paper (Bu et al. JBC 2011)

110 "kinase SPAs" appears to be an unusual expression and in the introduction the conclusion of the paper should not be foreclosed

114 The claim that SPA1 is necessary for light induced phosphorylation of PIF1 is still not backed by data

Fig 2c: I am still not convinced but additionally I wonder why the radiogram was shrank and duplicated? Why is there actually no loading control in all phostag gels?

195 recruitment does not necessarily mean SPA is the kinase

Fig 5c now the labelling is wrong, there is now no difference between lanes 3+4 and 5+6 – I do not really see why this figure was changed at all?

303 phyBdeltaC does not bind SPA1 and does not bind PIF1 – not necessarily proof for SPA1 light dependent kinase activity (Qiu et al. 2017).

515 "essential" kinases?

In figure legends and in the M&M section the space between number and unit is sometimes missing.

The M&M section was only minimally improved, only the examples I mentioned were changed.

They meant to be examples. Like the ones following here:

529 – "...otherwise as indicated." - makes no sense. Additionally, all mutants used should be named with reference here.

- missing articles at many places

- wrong use of plural and singular at many places

- "Proteins were induced" – no, expression was induced...

- "colony is inoculated..... "

- "collected on test tubes"

....and many more problems which could have been easily corrected by native speaker.

Reviewer #3 (Remarks to the Author):

I was asked to comment on whether the concerns of the 2nd round of review have been satisfactorily addressed and whether any concerns cast doubts about the conclusions of the paper.

I agree with the other two reviewers that demonstrating SPA1's kinase activity and its role in phytochrome signaling is a novel and important contribution to the plant photobiology field. The authors present convincing in vitro evidence for SPA1's kinase activity on PIF1. A large portion of the questions was concerning the interpretations of the biochemical and genetic evidence regarding the in vivo roles of the kinase activity of SPA1 as well as how SPA1 activity is regulated by phyB.

In particular, reviewer #2 raised a valid concern regarding whether “phyB-GFP Δ C is specifically defective in SPA1 interaction and the phyB-SPA1 interaction is necessary for the light-induced degradation of PIF1 in Arabidopsis”

I share the same concerns. Because phyB-SPA1 interaction is relevant to a number of conclusions, I would recommend the authors to (1) determine the subdomains of phyB sufficient for SPA1 interaction (or to verify whether the HKRD is sufficient to interact with phyB) and (2) to verify whether phyB Δ C is defective in HKRD dimerization. Here are the reasons behind these recommendations:

The current data cannot discern whether the defects of phyB- Δ C (removal of the C-terminal 80 amino acids) was specifically in SPA1-binding or due to large structural changes, for example, in HKRD dimerization. First, removal of the C-terminal 80 amino acids of phyB deletes a significant portion of the HKRD domain, including the F and G2 boxes. The schematic illustrations of the phyB- Δ C and phyB-28 in fig. 4b and fig. 7a are inaccurate. They should be corrected to reflect the truncated HKRD in phyB- Δ C and phyB-28.

Qiu et al. (Nat Commun 2017, 8:1905) showed that HKRD is the main dimerization domain for PHYB. The ATPase subdomain at the C-terminal end of HKRD, including the F and G2 boxes, are required for dimerization. Based on these published results, one would expect that phyB- Δ C cannot dimerize at the HKRD region. Therefore, the defect of phyB- Δ C in SPA1 binding could be due to a defect in dimerization of the C-terminal module of PHYB as opposed to lack of SPA1 interaction per se. If this is the case, SPA1 could interact with phyB through a region other than the HKRD but merely requires it in a dimer form.

The fact that phyB- Δ C can still interact with PIF1 in a light-dependent manner (Fig. 1f) does not prove that integrity of the HKRD region of phyB- Δ C is intact, because the light-dependent phyB-PIF1 interaction is mediated by the N-terminal module of phyB and there is a separate weaker dimerization domain at the GAF domain in the N-terminal module (Nat Commun 2017, 8:1905).

Defects in HKRD dimerization block phyB nuclear accumulation and photobody formation, and therefore, can almost abolish the function phyB (Nat Commun 2017, 8:1905). Krall et al. (PNAS 2000, 97:8169) showed that phyB-28 (deletion of the entire ATPase subdomain of HKRD) is a weak allele and phyB-28 can localize to the nucleus and form photobodies (a phenotype largely different from phyB- Δ C) (Chen et al. PNAS 2003, 100:14493). In contrast, a single amino acid substitution in the HKRD, D1040V, which abolishes HKRD dimerization, blocks phyB from localizing to the nucleus and to photobodies (similar to phyB- Δ C) (Nat Commun 2017, 8:1905). Therefore, the results that phyB- Δ C-GFP is defective in nuclear accumulation and photobody formation are consistent with the idea that phyB- Δ C is impaired in HKRD dimerization.

To mitigate these concerns, the authors should provide evidence showing that the C-terminal domain alone of phyB- Δ C can still dimerize or interact with itself (the data in Fig. S8b are unclear to draw a conclusion).

Other comments:

1. PHYB-SPA1 interaction is a major conclusion of the manuscript, I suggest to move the data showing the interactions of SPA1 to phyB-N and phyB-C (Figure S4B) to the main figures. These results show that the C-terminal module of PHYB is sufficient for SPA1 interaction. It would be nice if the authors could show further evidence that the HKRD is sufficient for SPA1 interaction.

2. In Figure 2d, it is puzzling why the phosphorylated band (band shift) was not detected for endogenous PIF1. It would be more convincing to show that phosphorylation of endogenous PIF1 is abolished in the spaQ mutant.

3. Line 201: Fig. S3c, the conclusion that SPA-dependent PIF1 phosphorylation is COP1-independent is based on data from *cop1-4*, a weak allele of COP1. It should be clarified that these results are not sufficient to conclude that PIF1 phosphorylation is COP1-independent.

Response to reviewer's comments

MS ID#: NCOMMS-18-33016B

MS TITLE: A phyB-PIF1-SPA1 kinase regulatory complex promotes photomorphogenesis in Arabidopsis

AUTHORS: Inyup Paik, Fulu Chen, Vinh Ngoc Pham, Ling Zhu, Jeong-Il Kim and Enamul Huq

Reviewers' comments:

Reviewer #1 (Remarks to the Author):

The manuscript has now been changed in some places to soften the previous hard claims of the authors. Nevertheless, several points still need attention and I do not agree on the authors disfavor for the scaffold hypothesis. I also think that the dark activity contradiction should be mentioned honestly –but agree with the authors that this is further project.

Author response: We now fully agree with the reviewer's suggestion on the importance of the scaffold hypothesis. Our SPA1 kinase mutant has reduced kinase activity, however, as it was suggested by a recent publication, the missense mutation might also disrupt the structural integrity of SPA1. We believe that the scaffold hypothesis and the kinase hypothesis are not mutually exclusive. Rather, those two are pointing to the same direction that the conserved amino acid in the kinase like domain might be necessary for both the structural integrity and the kinase activity. Therefore, we added following description in the discussion section of the manuscript.

“Moreover, the kinase like domain of SPA1 may act as a molecular scaffold for potential protein-protein interaction. The arginine 517 in SPA1 is very well conserved among many plant species, suggesting an importance of the structural integrity of the kinase domain (Fig. S1). It is possible that the SPA1 (R517E) missense mutation compromised not only the kinase activity, but

also the structural integrity of the SPA1 kinase domain, resulting in a defect in PIF1 degradation and seed germination in response to light. The biochemical basis for the failure of the SPA1 (R517E) missense mutation to rescue the *spaQ* seedling de-etiolation phenotype in the dark is still unknown. However, our hypothesis is that the kinase activity by itself and/or the structural information included within the kinase domain contribute to regulating seedling de-etiolation phenotypes.”

Further points (no specific order):

In the summary and as a heading the authors state

31 phyB interacts with SPA1 through its C-terminus and enhances the recruitment of PIF1 for phosphorylation

To my understanding SPA1 enhances the recruitment of PIF1 to phyB for phosphorylation?

Author response: In our opinion, the reviewer’s suggestion and our statement are both true. This is because all three proteins (phyB, SPA1, PIF1) can interact with each other. In that sense, any of the three proteins could serve as a molecular glue that enhances the interaction of the other two proteins. In this manuscript, however, our discussion mainly focused on the phyB-SPA1 interaction and the phosphorylation of PIF1 by SPA1. Thus, we would like to stay in the current statement that gives more emphasis on the phyB functions as a glue in the PIF1-SPA1 kinase interaction.

108 "no PIF kinase known" contradicts the senior author's own paper (Bu et al. JBC 2011)

Author response: Thanks for the suggestion. We previously reported that CK2 phosphorylates multiple ser/thr residues on PIF1 *in vitro*. This phosphorylation is not light-regulated (Bu et al. JBC 2011). We have discussed this in the discussion section that PIF1 phosphorylation by CK2 is light-independent. To avoid misinterpretation of the sentence, we have changed the original statement.

“Casein Kinase 2 (CK2) was previously shown to phosphorylate PIF1 *in vitro*, however, was not involved in the light-induced phosphorylation of PIF1. Therefore, the protein kinase necessary for the rapid light-induced PIF1 phosphorylation has not been identified”

110 "kinase SPAs" appears to be an unusual expression and in the introduction the conclusion of the paper should not be foreclosed

Author response: Thanks for the suggestion. We have corrected the sentence into this.

“Similar to the mammalian COP1-associated pseudo-kinase, Trib, plant COP1-associated SPA kinases have no known substrate,”

114 The claim that SPA1 is necessary for light induced phosphorylation of PIF1 is still not backed by data

Author response: The claim that SPA1 is necessary for light-induced phosphorylation of PIF1 is based on multiple independent evidence: i) SPA1 can directly phosphorylate PIF1 *in vitro*, ii) overexpression of *SPA1* results in faster phosphorylation than control *in vivo*, iii) light-induced phosphorylation is eliminated in *spaQ* background, iv) light-induced ubiquitination and degradation is reduced or absent in *spaQ* background, v) the kinase domain of SPA1 behaves like a kinase as supported by ATP binding, ATP-dependent phosphorylation assays, kinetic analysis, concentration-dependent phosphorylation of the substrate, and acid-base sensitivity assays, and vi) mutation in a critical residue in the kinase domain of SPA1 results in reduced phosphorylation *in vitro* and *in vivo*, and reduced degradation of PIF1 under light *in vivo*. These data strongly suggest that SPA1 is necessary for the light-induced phosphorylation of PIF1. However, we agree with the reviewer that our data does not support the claim that SPA1 is **sufficient** for light-induced phosphorylation of PIF1 as we could not show the light-induced phosphorylation *in vitro*.

Fig 2c: I am still not convinced but additionally I wonder why the radiogram was shrank and duplicated? Why is there actually no loading control in all phostag gels?

Author response: We apologize for this. It was our mistake during figure preparation. During our high resolution image preparation, the image was somehow shrank and duplicated. We have corrected the image. For phostag-gel, we performed the immunoprecipitation of TAP-PIF1 under different light conditions described and separate them on the phostag gel. This is why there is no RPT5 control available. The detailed procedure is described in the method section.

195 recruitment does not necessarily mean SPA is the kinase

Author response: We agree with the reviewer that recruitment does not mean SPA is the kinase. However, our claim that SPA1 is a kinase is based on multiple independent evidence: i) SPA1 can directly phosphorylate PIF1 *in vitro*, ii) overexpression of *SPA1* results in faster phosphorylation than control *in vivo*, iii) light-induced phosphorylation is eliminated in *spaQ* background, iv) light-induced ubiquitination and degradation is reduced or absent in *spaQ* background, v) the kinase domain of SPA1 behaves like a kinase as supported by ATP binding, ATP-dependent phosphorylation assays, kinetic analysis, concentration-dependent phosphorylation of the substrate, and acid-base sensitivity assays, and vi) mutation in a critical residue in the kinase domain of SPA1 results in reduced phosphorylation *in vitro* and *in vivo*, and reduced degradation of PIF1 under light *in vivo*. These data strongly suggest that SPA1 is a kinase for PIF1.

Fig 5c now the labelling is wrong, there is now no difference between lanes 3+4 and 5+6 – I do not really see why this figure was changed at all?

Author response: We apologize for this. We corrected the lane legends in lane 5 and 6. These two lanes only contain phyB as a kinase.

303 phyBdeltaC does not bind SPA1 and does not bind PIF1 – not necessarily proof for SPA1 light dependent kinase activity (Qiu et al. 2017).

Author response: As shown in our experiments, both phyB FL and phyB ΔC can bind to PIF1 normally *in vivo* (Fig. S8a) and *in vitro* (Fig. 4h).

515 "essential" kinases?

Author response: Thanks for the suggestion. We have deleted the word “essential”.

In figure legends and in the M&M section the space between number and unit is sometimes missing.

The M&M section was only minimally improved, only the examples I mentioned were changed.

They meant to be examples. Like the ones following here:

529 – "...otherwise as indicated." - makes no sense. Additionally, all mutants used should be named with reference here.

- missing articles at many places

- wrong use of plural and singular at many places

- "Proteins were induced" – no, expression was induced...

- "colony is inoculated..... "

- “collected on test tubes”

...and many more problems which could have been easily corrected by native speaker.

Author response: Thanks for the suggestion. We now have corrected the entire methods and materials section. We also added an additional table indicating the references for all the mutant and transgenic lines used in this study.

Reviewer #3 (Remarks to the Author):

I was asked to comment on whether the concerns of the 2nd round of review have been satisfactorily addressed and whether any concerns cast doubts about the conclusions of the paper.

Author response: We really appreciate that you agreed to review our revised manuscript.

I agree with the other two reviewers that demonstrating SPA1's kinase activity and its role in phytochrome signaling is a novel and important contribution to the plant photobiology field. The authors present convincing *in vitro* evidence for SPA1's kinase activity on PIF1. A large portion of the questions was concerning the interpretations of the biochemical and genetic evidence regarding the *in vivo* roles of the kinase activity of SPA1 as well as how SPA1 activity is regulated by phyB.

In particular, reviewer #2 raised a valid concern regarding whether “phyB-GFP Δ C is specifically defective in SPA1 interaction and the phyB-SPA1 interaction is necessary for the light-induced degradation of PIF1 in *Arabidopsis*”

I share the same concerns. Because phyB-SPA1 interaction is relevant to a number of conclusions, I would recommend the authors to (1) determine the subdomains of phyB sufficient for SPA1 interaction (or to verify whether the HKRD is sufficient to interact with phyB) and (2) to verify whether phyB Δ C is defective in HKRD dimerization. Here are the reasons behind these recommendations:

The current data cannot discern whether the defects of phyB- Δ C (removal of the C-terminal 80 amino acids) was specifically in SPA1-binding or due to large structural changes, for example, in HKRD dimerization. First, removal of the C-terminal 80 amino acids of phyB deletes a significant portion of the HKRD domain, including the F and G2 boxes. The schematic illustrations of the phyB- Δ C and phyB-28 in fig. 4b and fig. 7a are inaccurate. They should be corrected to reflect the truncated HKRD in phyB- Δ C and phyB-28.

Qiu et al. (Nat Commun 2017, 8:1905) showed that HKRD is the main dimerization domain for PHYB. The ATPase subdomain at the C-terminal end of HKRD, including the F and G2 boxes, are required for dimerization. Based on these published results, one would expect that phyB- Δ C cannot dimerize at the HKRD region. Therefore, the defect of phyB- Δ C in SPA1 binding could be due to a defect in dimerization of the C-terminal module of PHYB as opposed to lack of

SPA1 interaction per se. If this is the case, SPA1 could interact with phyB through a region other than the HKRD but merely requires it in a dimer form.

The fact that phyB- Δ C can still interact with PIF1 in a light-dependent manner (Fig. 1f) does not prove that integrity of the HKRD region of phyB- Δ C is intact, because the light-dependent phyB-PIF1 interaction is mediated by the N-terminal module of phyB and there is a separate weaker dimerization domain at the GAF domain in the N-terminal module (Nat Commun 2017, 8:1905).

Defects in HKRD dimerization block phyB nuclear accumulation and photobody formation, and therefore, can almost abolish the function phyB (Nat Commun 2017, 8:1905). Krall et al. (PNAS 2000, 97:8169) showed that phyB-28 (deletion of the entire ATPase subdomain of HKRD) is a weak allele and phyB-28 can localize to the nucleus and form photobodies (a phenotype largely different from phyB- Δ C) (Chen et al. PNAS 2003, 100:14493). In contrast, a single amino acid substitution in the HKRD, D1040V, which abolishes HKRD dimerization, blocks phyB from localizing to the nucleus and to photobodies (similar to phyB- Δ C) (Nat Commun 2017, 8:1905). Therefore, the results that phyB- Δ C-GFP is defective in nuclear accumulation and photobody formation are consistent with the idea that phyB- Δ C is impaired in HKRD dimerization.

To mitigate these concerns, the authors should provide evidence showing that the C-terminal domain alone of phyB- Δ C can still dimerize or interact with itself (the data in Fig. S8b are unclear to draw a conclusion).

Author response: Thank you for the detailed explanation about your concerns. We fully agree with the reviewer's concerns and suggestions. To investigate whether the isolated C-terminal domain of phyB (PHYB CT) and PHYB CT- Δ C can dimerize, we performed yeast-two-hybrid assays (LexA system) as the reviewer suggested. In our assays, we first found that the full-length C-terminal domain of phyB (PHYB CT) can dimerize efficiently with itself in yeast as reported (Qiu et. al., 2017). The PHYB CT (BD)-PHYB CT- Δ C(AD) also dimerized similar to the PHYB CT (BD)-PHYB CT(AD) combination. However, phyB CT- Δ C-(BD)-phyB CT- Δ C(AD) combination showed no significant difference compared to the empty vector control (Fig S8c),

suggesting that, as this reviewer predicted, the phyB CT- Δ C is compromised in dimerization. Based on these data, we modified the result section as following:

“However, the isolated phyB CT Δ C failed to dimerize in yeast assays, despite similar dimerization activity of the phyB CT Δ C with phyB CT in yeast two hybrid assays (Fig. S8c), suggesting that the dimerization might be defective in the isolated phyB CT Δ C, but not in the phyB-GFP Δ C context.”

While these data are important, our main conclusions are based on the phyB-GFP Δ C in the full-length context, not in the isolated C-terminal context. Our conclusions are based on:

- i) In native gel, phyB- Δ C holoprotein runs as a dimer similar to the full-length phyB (Fig. S8b).
- ii) The full-length phyB and phyB- Δ C interact with and sequester PIF1 in a similar manner in a light-dependent manner *in vitro* and *in vivo* (Fig. 4h, S8a, d), although this could be due to the interaction through the N-terminal half of phyB.
- iii) The phyB-GFP Δ C is largely functional in complementing the *phyB* mutant phenotype with very weak seedling de-etiolation phenotypes compared to full-length phyB-GFP (Fig. S7). phyB-GFP Δ C failed to induce phosphorylation and degradation of PIF1, resulting in reduced seed germination in a light-dependent manner (Fig. 6). In a previous study, Liu et al showed that monomeric phyB with an artificial dimerization and nuclear localization signal failed to rescue *phyB* mutant phenotype (Liu et al., 2012, Plant J 75, 915). Thus, the lack of dimerization can't explain the phyB-GFP Δ C phenotypes.
- iv) We also want to point out that the NG-GUS-NLS (Matsushita et al., Nature 2003), in which the C-terminal half of the phyB is deleted and replaced with an obligatory dimerization domain GUS, has significantly reduced degradation of PIF1 (Fig. S8e)

and PIF3 (Park et al., 2012, Plant J; Park et al., 2018 Plant Cell). This is a circumstantial evidence that the dimerization defect is not a major cause of the abolished PIF1 phosphorylation and degradation in the phyB- Δ C. It is the C-terminal half of phyB that induces degradation of PIF1 and PIF3 as has been shown by Qiu et al., 2017, Nature Comm. Our hypothesis is that the C-terminal domain recruits SPA1 to induce phosphorylation and degradation of PIF1. In support of this hypothesis, we now show that the last 85 amino acids of phyB are sufficient for interaction with SPA1 (Fig. 4c).

- v) The phyB- Δ C still contains the essential amino acid for phyB dimerization (D1040) (Qiu et al., 2017). Thus, our results suggest the phyB- Δ C still can dimerize in the full-length context. We cannot completely rule out the possibility that the phyB- Δ C might still have an additional defect other than SPA1 interaction.

Because of the uncertainty, we modified the discussion section as follows:

“However, it is still possible that phyB- Δ C might have additional intrinsic defects that were not fully addressed in our study. For example, the phyB-GFP Δ C exhibits defect in the photobody formation similar to the PBY18-YFP (D1040V), which cannot dimerize (Qiu et al., 2017). But unlike PBY18-YFP (D1040V) which lacks PIF3 interaction, phyB-GFP Δ C still interacts with PIF1 *in vitro* and *in vivo*. Because the isolated phyB CT Δ C failed to dimerize in yeast, it is still possible that the last 80 amino acids of phyB contributes to the dimerization and photobody formation of phyB-GFP Δ C.”

Other comments:

1. PHYB-SPA1 interaction is a major conclusion of the manuscript, I suggest to move the data showing the interactions of SPA1 to phyB-N and phyB-C (Figure S4B) to the main figures.

These results show that the C-terminal module of PHYB is sufficient for SPA1 interaction. It would be nice if the authors could show further evidence that the HKRD is sufficient for SPA1 interaction.

Author response: Thanks for the suggestion. We also performed yeast-two-hybrid assays to investigate if the HKRD domain is sufficient for SPA1 interaction. We cloned two additional vectors that harbor fragments of HKRD, LexA(BD)-CA3 (Qiu et al., 2017) and LexA-C (1079-End). Both fragments showed comparable interaction with the full-length SPA1 in pJG4-5 (AD vector), suggesting that our mapped SPA1 interaction domain in phyB is sufficient for SPA1 interaction in yeast. We also added following description in the results section.

And in the results section:

“Conversely, we also examined whether the potential SPA1 interaction domain in phyB is sufficient for the interaction with SPA1 in yeast two hybrid assays. Results show that the extreme C-terminal (C) domain was able to interact with SPA1 and the interaction level was comparable to the full-length C-terminal PHYB (Fig. 4c).”

2. In Figure 2d, it is puzzling why the phosphorylated band (band shift) was not detected for endogenous PIF1. It would be more convincing to show that phosphorylation of endogenous PIF1 is abolished in the *spaQ* mutant.

Author response: This is an important issue we have been dealing with for long time. In all our previous publications since 2008 (Shen et al., 2008), we repeatedly observed that our PIF1 antibody cannot detect phosphorylated form of endogenous PIF1 (shifted band). Our speculation is that the endogenous PIF1 is extremely unstable in its phosphorylated form and goes through 26S proteasome-mediated degradation quickly beyond the detection limit. Therefore, we are unable to detect the phosphorylated form of endogenous PIF1. Instead, we have shown in multiple ways that TAP-PIF1 is functionally close to the endogenous PIF1 (Shen et al., 2008), and its' phosphorylation is abolished in the *spaQ* mutant. We hope this will satisfy the reviewer's concern.

3. Line 201: Fig. S3c, the conclusion that SPA-dependent PIF1 phosphorylation is COP1-independent is based on data from *cop1-4*, a weak allele of COP1. It should be clarified that these results are not sufficient to conclude that PIF1 phosphorylation is COP1-independent.

Author response: Thanks for the suggestion. We fully agree that our conclusion solely based on the result from *cop1-4*, a weak allele of COP1, in which COP1 is partially functioning in unknown ways, is premature. We have added the following sentence to clarify that our current conclusion cannot rule out the potential function of COP1 in PIF1 phosphorylation.

“Since *cop1-4* is a weak allele and still maintains some COP1 activity (McNellis et al., 1994), further experiments are necessary to exclude the role of COP1 in PIF1 phosphorylation.”

REVIEWERS' COMMENTS:

Reviewer #3 (Remarks to the Author):

The authors have carefully addressed my questions and performed new experiments to further define the interaction between SPA1 and PHYB's C-terminal module. The new results provide evidence supporting the conclusion that SPA1 interacts specifically with the very C-terminal segment of PHYB. I only have a few minor comments/suggestions regarding this particular conclusion.

1. Line 239: "Results show that the extreme C-terminal (C) domain was able to interact with SPA1 and the interaction level was comparable to the full-length C-terminal PHYB (Fig. 4c)."

Comparing the interaction data relative to the GAD controls in Fig. 4c with those in Fig. 4a, the interaction results between SPA1 and the PHYB-CT were inconsistent: in Fig. 4a the relative interaction activity is about 8-fold, whereas in Fig. 4c it is only about 2-fold. Please explain the discrepancies.

2. Line 241-243: If B-C is both necessary and sufficient for SPA1 interaction, because B-C itself cannot dimerize (Qiu et al 2017), these results suggest that SPA1 might not need to interact with a dimer form of the HKRD. The authors might want to discuss this possibility.

3. Line 229: "we narrowed down the SPA1 interaction domain to the last 80 amino acids of the C-terminal end of phyB as a deletion mutant lacking the last 80 amino acids from the C-terminal half of phyB (PHYB CT DC) failed to interact with full-length SPA1 (Fig. 4a).

The data in Fig.4a suggest that the last 80 amino acids of PHYB's C-terminus is required for SPA1 interaction, but do not necessarily mean that this is the SPA1 interacting domain. Please rewrite this sentence to make this point more precisely.

4. Figure 4b, please explain what D153-PHYB stands for in the legend, and please make the name consistent throughout the figure.

5. Please use "phyB" or "PHYB" consistently throughout the text.

Response to reviewer's comments

MS ID#: NCOMMS-18-33016C

MS TITLE: A phyB-PIF1-SPA1 kinase regulatory complex promotes photomorphogenesis in Arabidopsis

AUTHORS: Inyup Paik, Fulu Chen, Vinh Ngoc Pham, Ling Zhu, Jeong-II Kim and Enamul Huq

REVIEWERS' COMMENTS:

Reviewer #3 (Remarks to the Author):

The authors have carefully addressed my questions and performed new experiments to further define the interaction between SPA1 and PHYB's C-terminal module. The new results provide evidence supporting the conclusion that SPA1 interacts specifically with the very C-terminal segment of PHYB. I only have a few minor comments/suggestions regarding this particular conclusion.

1. Line 239: "Results show that the extreme C-terminal (C) domain was able to interact with SPA1 and the interaction level was comparable to the full-length C-terminal PHYB (Fig. 4c)."

Comparing the interaction data relative to the GAD controls in Fig. 4c with those in Fig. 4a, the interaction results between SPA1 and the PHYB-CT were inconsistent: in Fig. 4a the relative interaction activity is about 8-fold, whereas in Fig. 4c it is only about 2-fold. Please explain the discrepancies.

Author response: The quantitative beta-galactosidase activity can vary batch to batch even in the same lab depending on the transformation efficiency and the expression of the fusion protein in yeast. This is why we included negative and positive controls in each of the quantitative yeast two hybrid assays. In this case, two experiments were performed independently by two different researchers and the results showed fold difference between the two experiments, but the overall data support the same conclusion. Since the activity showed comparable value within the same batch of transformation and quantitation, we believe that these results are reliable.

2. Line 241-243: If B-C is both necessary and sufficient for SPA1 interaction, because B-C itself cannot dimerize (Qiu et al 2017), these results suggest that SPA1 might not need to interact with a dimer form of the HKRD. The authors might want to discuss this possibility.

Author response: The reviewer's suggestion might be true. However, phytochromes are functional only in their dimer forms and exist as dimers in vivo. SPA proteins might also exist as

dimers and each dimer can interact with another dimer. However, we do not have any stoichiometric data to support these claims. Thus, we prefer not to discuss these details at this point.

3. Line 229: “we narrowed down the SPA1 interaction domain to the last 80 amino acids of the C-terminal end of phyB as a deletion mutant lacking the last 80 amino acids from the C-terminal half of phyB (PHYB CT DC) failed to interact with full-length SPA1 (Fig. 4a).

The data in Fig.4a suggest that the last 80 amino acids of PHYB’s C-terminus is required for SPA1 interaction, but do not necessarily mean that this is the SPA1 interacting domain. Please rewrite this sentence to make this point more precisely.

Author response: Thanks for the suggestion. We corrected the sentence into this

“Further deletion analysis showed that the last 80 amino acids of the C-terminal end of phyB is required for the interaction with full-length SPA1 (Fig. 4a).”

4. Figure 4b, please explain what D153-PHYB stands for in the legend, and please make the name consistent throughout the figure.

Author response: We added a reference and proper description in the legend.

5. Please use “phyB” or “PHYB” consistently throughout the text.

Author response: We followed the nomenclature published by fellow scientists. (1994 Quail et al.) The “phyB” was used to describe a holo-protein while “PHYB” was used to describe an apo-protein. We reviewed the manuscript one more time to correct usage of the nomenclature.